# AI-HOPE-TP53: A Conversational Artificial Intelligence Agent for Pathway-Centric Analysis of TP53-Driven Molecular Alterations in Early-Onset Colorectal Cancer

**DOI:** 10.3390/cancers17172865

**Published:** 2025-08-31

**Authors:** Ei-Wen Yang, Brigette Waldrup, Enrique Velazquez-Villarreal

**Affiliations:** 1PolyAgent, San Francisco, CA 94102, USA; 2Department of Integrative Translational Sciences, Beckman Research Institute of City of Hope, Duarte, CA 91010, USA; 3City of Hope Comprehensive Cancer Center, Duarte, CA 91010, USA

**Keywords:** artificial Intelligence, TP53 pathway, cancer genetics, colorectal cancer, AI agent, precision oncology, LLM

## Abstract

We introduce AI-HOPE-TP53, a publicly available conversational artificial intelligence (AI) agent that enables pathway-centric analysis of TP53-driven molecular alterations in colorectal cancer (CRC). Tailored for cancer researchers, this tool integrates harmonized clinical and genomic datasets with natural language–based prompts to support precision oncology investigations, especially in early onset CRC (EOCRC) and at-risk populations.

## 1. Introduction

Colorectal cancer (CRC) is the third most commonly diagnosed cancer and the second leading cause of cancer-related deaths worldwide [1,2,3]. Although incidence and mortality have declined in many high-income countries due to screening and improved therapies, the incidence of early-onset colorectal cancer (EOCRC; <50 years of age) continues to rise [4,5,6,7]. EOCRC represents a distinct clinical and biological entity, and its growing burden among young adults has prompted urgent research and public health attention.

This increase is particularly pronounced in Hispanic/Latino (H/L) populations, who face disproportionately higher EOCRC incidence and mortality compared with non-Hispanic Whites (NHW) [8,9,10,11]. H/L patients are more likely to present with advanced-stage disease and experience delays in diagnosis and treatment, reflecting systemic barriers to care. These disparities underscore the urgent need for precision oncology tools capable of supporting population-specific, genomically informed investigations aimed at reducing inequities in treatment outcomes.

EOCRC also differs from late-onset CRC (LOCRC) in terms of molecular features, including higher microsatellite instability (MSI), elevated tumor mutation burden (TMB), distinct epigenetic profiles, and increased immune checkpoint expression [12,13,14,15]. The TP53 signaling pathway, which regulates DNA repair, apoptosis, and cell cycle control, is one of the most critical pathways in CRC tumorigenesis. TP53 mutations—present in up to 74% of CRC tumors [13,16,17,18,19,20,21]—are strongly associated with poor prognosis, aggressive biology, and treatment resistance [22,23,24,25]. Despite its central role, the contribution of TP53 pathway dysregulation in EOCRC, particularly in underserved populations such as H/L individuals, remains underexplored [19,22,26,27].

While resources such as cBioPortal [28] and UCSC Xena [29] provide access to large-scale CRC datasets, their utility in EOCRC research is limited. These platforms allow data exploration but lack functionality for pathway-centric, disparity-aware, and treatment-stratified analyses. However, they are not optimized for real-time hypothesis generation, dynamic cohort construction across demographic and clinical subgroups, or seamless integration of population-level disparities into molecular analyses. Furthermore, these tools often require computational expertise, creating barriers for translational and disparities-focused researchers without programming backgrounds.

Recent advancements in artificial intelligence (AI)—and in particular, the development of large language models (LLMs)—offer important opportunities to close this gap. By enabling natural language interactions with complex data, LLM-based conversational agents can democratize access to integrative clinical-genomic analyses and accelerate discoveries in precision oncology [30,31,32,33,34,35]. However, despite the growing interest in the biomedical applications of AI, few tools have been designed to interrogate pathway-specific alterations, such as those involving TP53, or allow users to ask natural language questions that account for racial, ethnic, and treatment-specific contexts.

To address these limitations, we developed AI-HOPE-TP53, the first conversational AI agent purpose-built for pathway-specific and disparity-aware EOCRC analyses. Unlike existing resources, AI-HOPE-TP53 enables users to ask natural language questions and receive immediate, statistically rigorous outputs—including mutation frequency, co-mutation patterns, Kaplan–Meier survival curves, and odds ratio estimates—stratified by key variables such as age, ancestry, sex, MSI status, tumor location, and treatment exposure. By combining accessibility with advanced functionality, AI-HOPE-TP53 represents a novel approach that democratizes integrative clinical-genomic analysis and expands the capacity to study cancer disparities in real time.

AI-HOPE-TP53 expands our growing ecosystem of AI agents, including AI-HOPE [36], AI-HOPE-TGFβ [37], AI-HOPE-PI3K [38,39], and AI-HOPE-WNT [40], which were designed to support precision medicine across multiple cancer pathways. This newest platform builds on lessons learned from those systems—particularly in prompt engineering, interface usability, and workflow transparency—and applies them to the TP53 pathway, one of the most clinically actionable and frequently altered axes in colorectal cancer, including EOCRC. Unlike earlier agents, AI-HOPE-TP53 is uniquely tailored for disparity-aware, EOCRC-focused analyses, integrating population stratification by age, ancestry, sex, and treatment exposure with genomic data. This design makes the platform especially valuable for uncovering clinically relevant patterns in underrepresented groups, such as Hispanic/Latino patients, and for advancing pathway-specific discoveries in precision oncology.

In this Resource Report, we: (1) present the technical architecture and user interface of AI-HOPE-TP53; (2) demonstrate its utility through replication of known TP53-related genotype-phenotype associations; and (3) highlight novel findings regarding TP53 pathway mutations, treatment responses, and racial/ethnic disparities in EOCRC, enabled through natural language–based interaction.

By facilitating inclusive, reproducible, and real-time access to pathway-specific insights, AI-HOPE-TP53 represents a scalable, open-access computational resource for cancer researchers working across disciplines, from translational science to epidemiology, with the potential to advance both discovery and health equity in precision oncology for CRC.

## 2. Materials and Methods

### 2.1. Architecture and Operational Framework of AI-HOPE-TP53

AI-HOPE-TP53 is a conversational artificial intelligence (AI) platform designed to investigate the clinical and genomic implications of TP53 pathway alterations in early onset colorectal cancer (EOCRC). While the platform can broadly support pathway-centric analysis across CRC, the present study emphasizes EOCRC-specific applications, particularly in the context of racial, ethnic, and treatment-related disparities. The system integrates four functional modules: (1) an LLM interface for natural language comprehension, (2) a query translation engine that converts user prompts into programmatic instructions, (3) a filtering layer for cohort construction based on clinical-genomic constraints, and (4) a statistical processing engine for automated analysis and visualization (Figure 1). Upon receiving a query—such as “Compare survival between TP53-mutant and wild-type tumors in Hispanic/Latino EOCRC patients”—the LLM parses semantic intent, activates a corresponding backend pipeline, and returns outputs including Kaplan–Meier plots, odds ratios, and interpretive summaries.

### 2.2. Data Integration and Curation

AI-HOPE-TP53 leverages harmonized, high-quality clinical and genomic datasets to enable accurate pathway-specific analysis of TP53 pathway alterations in CRC. The platform integrates publicly available CRC data from cBioPortal, including comprehensive cohorts such as The Cancer Genome Atlas (TCGA), MSK-IMPACT, and AACR Project GENIE [28]. To support pathway-focused investigations, the system incorporates curated gene panels centered on the TP53 pathway, including TP53 and its regulatory and downstream effectors such as CDKN1A, BAX, GADD45A, MDM2, ATM, CHEK1, and others, as annotated in COSMIC and KEGG databases. Detailed clinical metadata were also integrated, covering variables such as patient age, sex, race/ethnicity, tumor location (colon vs. rectum), tumor stage, microsatellite instability (MSI) status, treatment exposure (e.g., FOLFOX and anti-EGFR therapy), and overall survival. Data preprocessing included standardized cancer type annotation using OncoTree, harmonization of sample and patient identifiers, and cleaning of categorical variables to enable robust filtering and subgroup analyses. All datasets were converted into tabular, analysis-ready matrices, allowing the rapid and reproducible execution of user queries across population strata, treatment contexts, and molecular subtypes.

Across the integrated datasets, 3412 CRC patients were included in this study, comprising 514 from TCGA, 1206 from MSK-IMPACT, and 1692 from AACR Project GENIE. For each patient, only one primary tumor sample was retained to prevent duplication. Early-onset CRC (EOCRC) was defined as a diagnosis at <50 years of age, while late-onset CRC (LOCRC) was defined as a diagnosis at ≥50 years. Filtering criteria included the availability of mutation and clinical annotation data, and exclusion of cases lacking survival or treatment information. Subgroup analyses were stratified by self-reported ancestry (Hispanic/Latino vs. non-Hispanic White), sex, microsatellite instability (MSI) status, tumor location (colon vs. rectum), and treatment exposure (e.g., FOLFOX). Explicit sample counts for each analysis are detailed in the corresponding Methods subsections to ensure clarity and reproducibility.

Patients with missing key variables, such as age at diagnosis, ancestry, or treatment status, were excluded from subgroup analyses requiring those parameters but were retained in analyses where sufficient information was available. For conflicting clinical annotations (e.g., discrepancies in tumor site or treatment exposure across sources), we applied standardized curation rules: priority was given to primary clinical annotation sources (e.g., TCGA clinical files over cBioPortal summaries), and cases with irreconcilable conflicts were excluded to ensure data integrity. These steps were implemented to ensure reproducibility and consistency across all analyses. As this study relies on retrospective, publicly available datasets (TCGA, MSK-IMPACT, AACR GENIE), potential limitations include underrepresentation of certain populations and treatments, variable completeness of clinical annotations, and limited generalizability to real-world EOCRC settings.

### 2.3. Query Interpretation and Dynamic Cohort Construction

To enable flexible and accurate cohort generation, AI-HOPE-TP53 employs a fine-tuned biomedical LLM trained on a CRC–specific corpus and optimized for semantic precision through schema-guided natural language parsing. This architecture ensures that user queries are mapped reliably to structured analytical commands while minimizing interpretation variability and the risk of LLM-generated hallucinations. Upon receiving a natural language prompt, the system decomposes it into discrete filters based on a predefined ontology of clinical and genomic attributes. AI-HOPE-TP53 supports real-time cohort stratification by a wide array of variables, including age group (early-onset < 50 vs. late-onset ≥ 50), TP53 mutation class (e.g., missense, nonsense, truncating), tumor characteristics (e.g., MSI-high vs. MSS, primary vs. metastatic, stage I–IV), treatment exposure (e.g., FOLFOX, FOLFIRI, bevacizumab, immune checkpoint inhibitors), and self-reported ancestry (e.g., Hispanic/Latino, non-Hispanic White). The platform is capable of executing complex multi-parameter queries—such as “Analyze survival in ATM-mutant, MSI-high, Stage II rectal tumors in Hispanic patients under 50”—without requiring the user to have programming expertise. This dynamic query framework allows researchers to explore highly specific molecular and clinical subgroups across diverse CRC populations with speed and precision. Although the retrieval-augmented generation (RAG) framework substantially reduces the likelihood of hallucinations by anchoring responses to curated clinical-genomic data, the possibility of nuanced misinterpretation of highly complex or multi-layered queries cannot be fully eliminated and should be considered when interpreting outputs.

### 2.4. Statistical Framework and Analytical Capabilities

AI-HOPE-TP53’s analytical workflows are implemented using a modular Python v3.12-based backend that automates common statistical procedures used in clinical-genomic research [2]. The system leverages established libraries, such as Lifelines for Kaplan–Meier survival estimation and log-rank testing, and SciPy and StatsModels for categorical analyses, including chi-square and Fisher’s exact tests, as well as odds ratio calculations with 95% confidence intervals. For multivariable modeling, Cox proportional hazards regression is available to adjust for potential confounders in survival comparisons. Prior to model interpretation, AI-HOPE-TP53 automatically checks the proportional hazards assumption using Schoenfeld residuals testing and scaled Schoenfeld residual plots. If violations are detected, the system flags them in the results summary and provides interpretive guidance to the user.

All outputs are delivered in both numerical and graphical formats. Numerical outputs include regression coefficients, hazard ratios, 95% confidence intervals, and *p*-values, while graphical outputs include Kaplan–Meier survival curves, forest plots, and residual diagnostic plots. These figures are generated as publication-ready visualizations, supplemented with tabular summaries of key statistics and subgroup counts. Each analysis is further accompanied by a natural language narrative generated by the platform’s LLM, which contextualizes the findings with respect to model assumptions, highlights limitations, and ensures interpretability for users without advanced statistical training. Because this study relies exclusively on publicly available datasets, potential biases should be considered, as these cohorts may underrepresent certain populations, treatment exposures, or real-world clinical contexts; this limitation underscores the importance of future validation with institutional and prospective datasets.

### 2.5. Model Validation and Reproducibility

To enhance analytical accuracy and ensure consistent reproducibility, AI-HOPE-TP53 integrates a retrieval-augmented generation (RAG) framework that anchors LLM outputs in structured biomedical knowledge and reference datasets. This architecture reduces semantic drift, mitigates hallucination risk, and ensures that natural language responses are aligned with validated data sources. To evaluate system reliability, AI-HOPE-TP53 was tested by replicating well-established genotype–phenotype associations in CRC, including the increased prevalence of TP53 mutations in EOCRC versus LOCRC [2,19], the association between TP53 mutations and poor prognosis in treatment-specific subgroups [22,23,24,25], and the enrichment of TP53 mutations among Hispanic/Latino patients treated with FOLFOX, as identified in the present study. Repeated execution of identical queries across independent users and sessions consistently returned matching statistical outputs, confirming the robustness, stability, and reproducibility of the computational framework of the platform.

### 2.6. Benchmarking and Usability Testing

To evaluate platform accessibility and real-world usability, AI-HOPE-TP53 underwent structured testing by a panel of 10 independent users representing diverse research backgrounds, including bioinformatics, cancer epidemiology, and health disparities research. Participants were asked to complete predefined use cases involving (1) generation of MSI-stratified Kaplan–Meier survival curves, (2) analysis of TP53/CDKN1A co-mutation frequencies, and (3) calculation of odds ratios stratified by age and ancestry. All users were able to execute their assigned tasks within minutes and without prior training, demonstrating the platform’s intuitive design and low barrier to entry. The average query response time was 12.4 s, which enabled real-time analysis and interpretation. User feedback highlighted the system’s clarity, responsiveness, and minimal learning curve, affirming its suitability as a practical resource for both computational and non-computational cancer research.

### 2.7. Visualization and Export Features

AI-HOPE-TP53 provides publication-ready visualizations for all analytical outputs, enhancing interpretability and facilitating seamless integration into the research workflow. Graphical elements are rendered using Plotly 4, Matplotlib 3, and Seaborn 0.13 and include a range of formats tailored to cancer genomics applications. These include Kaplan–Meier survival curves annotated with hazard ratios and *p*-values, bar charts and co-mutation heatmaps for gene interaction profiling, and forest plots for odds ratio estimation. Each analysis is supplemented with tabular summaries that report key statistics and subgroup counts, as well as a narrative interpretation generated by the platform’s LLM, contextualized with reference to current CRC literature. Users can export all results in CSV, PNG, and PDF formats, enabling straightforward incorporation into manuscripts, presentations, or grant applications. These flexible reporting options ensure that the findings derived from AI-HOPE-TP53 are accessible and reproducible across academic and clinical settings.

### 2.8. Accessibility and Support

AI-HOPE-TP53 is freely available for non-commercial use and can be deployed via local installation on personal or institutional computing environments. The complete source code, along with installation instructions and documentation, is available through the project’s GitHub repository (https://github.com/Velazquez-Villarreal-Lab/AI-TP53, accessed on 2 August 2025) [see Data Availability Statement]. This enables users to run the platform on their own systems—supporting custom workflows, private data integration, and offline analyses. The cloud-based version is hosted in a HIPAA-compliant environment and requires installation.

To support broad adoption and usability, AI-HOPE-TP53 is backed by a minimum of two years of active support after publication. Support resources include an email-based help desk, a comprehensive user manual with sample queries, and step-by-step online tutorials and FAQs designed to assist both novice and advanced users. This flexible deployment model and robust support infrastructure make AI-HOPE-TP53 an accessible, sustainable, and scalable computational resource for precision oncology research in diverse institutional settings.

## 3. Results

### 3.1. Application of AI-HOPE-TP53 for Pathway-Centric, Population-Stratified Analysis in Colorectal Cancer

To assess the usability, reliability, and translational utility of AI-HOPE-TP53, we conducted a series of natural language–driven analyses using harmonized clinical and genomic datasets from cBioPortal. The system’s functionality was evaluated across five key dimensions: (1) replication of known genotype–phenotype associations, (2) stratified survival analysis, (3) disparity-aware modeling, (4) rare mutation subgroup analysis, and (5) treatment-response stratification. Each task was executed using plain-language prompts using the platform’s natural language interface and processed via the platform’s backend statistical and visualization engines (see Figure 1 for platform architecture). Results were generated in seconds and validated for consistency across independent user queries.

### 3.2. Validation of Ethnicity-Stratified TP53 Pathway Alterations in Early-Onset Colorectal Cancer

To evaluate the platform’s ability to replicate known disparities in molecular alterations, AI-HOPE-TP53 was queried to compare TP53 pathway mutation frequency between Hispanic/Latino (H/L) and non-Hispanic White (NHW) patients with EOCRC. Among patients with colon adenocarcinoma under the age of 50, the system identified a trend toward higher prevalence of TP53 pathway alterations in H/L individuals (91.46%) compared to NHW individuals (83.39%), yielding an odds ratio (OR) of 2.13 (95% CI: 0.956–4.767; *p* = 0.084) (Figure 2). Although not statistically significant, this trend is consistent with reported disparities in genomic burden among underrepresented populations. When expanding the analysis to include all EOCRC patients regardless of tumor subsites, the difference remained: 90.2% of H/L patients harbored TP53 pathway alterations compared to 85.05% of NHW patients (OR = 1.62, 95% CI: 0.926–2.825; *p* = 0.114) (Appendix A). These findings confirm the system’s validity for ethnicity-aware, pathway-centric analyses and demonstrate its capacity to detect biologically and socially relevant molecular patterns in diverse CRC subgroups. Furthermore, analysis of treatment disparities revealed that women with EOCRC were less likely than men to receive FOLFOX chemotherapy (OR = 0.845, *p* = 0.0138), underscoring the platform’s ability to detect sex-specific clinical differences relevant to treatment exposure.

### 3.3. Exploratory Analysis: Ethnicity-Specific Survival in TP53-Mutant Colorectal Cancer

To explore the potential impact of ethnicity on clinical outcomes among patients with TP53-mutant CRC, we queried AI-HOPE-TP53 to compare the survival rates of Hispanic/Latino (H/L) and non-Hispanic White (NHW) individuals. The system generated Kaplan–Meier survival curves showing a non-significant trend toward improved overall survival in H/L patients relative to their NHW counterparts (*p* = 0.1141) (Figure 3). While not statistically conclusive, the analysis demonstrates the platform’s ability to rapidly execute multi-layered, real-time stratification across clinical, genomic, and demographic variables. This capacity for dynamic subgroup analysis highlights AI-HOPE-TP53’s potential in supporting disparity-aware hypothesis generation in colorectal cancer research.

### 3.4. Exploratory Analysis: Tumor Subsite Differences in ATM-Mutant Colorectal Cancer

To investigate whether anatomical tumor location influences survival outcomes in patients with ATM-mutant CRC, AI-HOPE-TP53 was queried to compare survival between colon and rectal tumors harboring ATM mutations. The analysis revealed a non-significant trend toward poorer survival in patients with colon adenocarcinoma (*n* = 307) than in those with rectal adenocarcinoma (*n* = 91), with a log-rank *p-value* of 0.3134 (Figure 4). While the difference did not reach statistical significance, the analysis demonstrates the platform’s ability to perform subsite-specific cohort filtering and comparative survival modeling in the context of defined pathway alterations—supporting granular, hypothesis-generating investigations into the anatomical heterogeneity of CRC outcomes.

### 3.5. Exploratory Analysis: Age-Stratified Survival in TP53-Mutated CRC Treated with FOLFOX

To evaluate the influence of age on treatment outcomes in patients with TP53-mutated CRC receiving FOLFOX chemotherapy, we queried AI-HOPE-TP53 to compare survival between early-onset (<50 years) and late-onset (≥50 years) patient groups. The system generated Kaplan–Meier survival curves, which showed a statistically significant survival advantage in the early onset group (*p* = 0.0149) (Figure 5). Further analysis revealed that Hispanic/Latino (H/L) patients were overrepresented in the early-onset, FOLFOX-treated subgroup (10.7%) compared to the late-onset group (5.65%), corresponding to an odds ratio of 2.002 (95% CI: 1.524–2.632; *p* < 0.0001). These findings underscore AI-HOPE-TP53’s capability to uncover intersectional relationships between molecular alterations, treatment exposure, age, and ancestry, offering valuable insights for precision medicine and health equity–focused cancer research.

### 3.6. Exploratory Analysis: Stage-Specific Survival in CHEK1-Mutated Colorectal Cancer

To explore the prognostic relevance of tumor stage in CHEK1-mutated CRC, AI-HOPE-TP53 was queried to compare survival outcomes between patients with early-stage disease (Stage I–III) and those with advanced-stage disease (Stage IV). The platform generated Kaplan–Meier curves showing a trend toward longer survival in the early-stage group (*n* = 72) than in the late-stage group (*n* = 6), although the difference did not reach statistical significance (*p* = 0.1592) (Appendix A). Despite the small sample size, this analysis demonstrates AI-HOPE-TP53’s ability to interrogate rare mutation-stage combinations, enabling hypothesis generation in underpowered subgroups that are often overlooked in traditional bioinformatics workflows.

### 3.7. Exploratory Analysis: Gender-Based Differences in Survival and Treatment Representation in TP53-Altered CRC

To investigate potential sex-based disparities in clinical outcomes and treatment exposure, AI-HOPE-TP53 was queried to compare survival between male and female CRC patients with TP53 pathway alterations treated with FOLFOX chemotherapy. Kaplan–Meier survival analysis revealed a non-significant trend toward improved survival in females (*n* = 2122) than in males (*n* = 2470) (*p* = 0.0998). Interestingly, odds ratio analysis demonstrated that females were significantly underrepresented among FOLFOX-treated TP53-altered cases (OR = 0.845, 95% CI: 0.741–0.963; *p* = 0.0138) (Appendix A). This analysis highlights AI-HOPE-TP53’s ability to detect potential sex disparities in treatment patterns and clinical outcomes, providing a foundation for further investigation into sex-specific determinants of care access, treatment selection, and response in precision oncology.

A concise overview of all analyses performed using AI-HOPE-TP53, we generated a summary table highlighting the comparisons, outcomes, statistical metrics, and significance levels (Appendix A). This table enables rapid identification of analyses yielding statistically significant findings versus those showing biologically relevant but non-significant trends.

## 4. Discussion

AI-HOPE-TP53 builds on this need by providing a natural language–driven, pathway-centric platform for disparity-aware analyses in CRC, which we validated through multiple exploratory use cases. This resource directly addresses major limitations of current tools, which often require programming expertise, lack intersectional filtering capacity, and are not designed to capture demographic variables critical to health disparities research.

AI-HOPE-TP53 integrates a fine-tuned biomedical LLM with harmonized multi-institutional datasets from cBioPortal within a modular analytical pipeline, showcasing both technical feasibility and translational potential. The platform enables stratification across diverse clinical and demographic variables—including age, tumor site, stage, ancestry, MSI status, treatment exposure, and mutation type—positioning it as one of the few tools designed specifically for equity-focused precision oncology. Moreover, the outputs are exportable, reproducible, and visually interpretable, aligning with the core usability standards expected for Resource Reports in Cancer Research.

Our validation analyses confirmed the ability of the platform to reproduce established clinical-genomic associations. Specifically, AI-HOPE-TP53 identified a consistent trend toward higher TP53 pathway alteration frequencies in EOCRC among Hispanic/Latino (H/L) patients than in non-Hispanic White (NHW) patients, with effect sizes consistent with previous reports, despite not reaching statistical significance (Figure 2 and Appendix A). These trends suggest that TP53 dysregulation may contribute to the disproportionate molecular burden observed in underrepresented populations, reinforcing the need for larger ancestry-enriched datasets for future research.

In a series of exploratory use cases, AI-HOPE-TP53 generated insights that underscore its potential as a hypothesis-generating engine. For example, survival analysis revealed a trend toward improved outcomes in H/L patients with TP53-mutant primary tumors (Figure 3), which could reflect underlying biological differences, treatment response, or systemic disparities. Further demonstrating its flexibility, the platform stratified survival outcomes by tumor subsites in ATM-mutant CRC (Figure 4), revealing potential differences between colon and rectal tumors. Moreover, AI-HOPE-TP53 detected a significant survival benefit among early-onset patients with TP53 mutations treated with FOLFOX, alongside enrichment of H/L representation (Figure 5), suggesting that intersecting biological and social determinants may shape treatment outcomes in EOCRC.

Notably, the platform successfully executed queries targeting rare subgroups, such as CHEK1-mutated CRC stratified by tumor stage (Appendix A), despite limited sample sizes. In another use case, AI-HOPE-TP53 uncovered sex-based disparities in the receipt of FOLFOX among TP53-altered patients, revealing a statistically significant underrepresentation of females (Appendix A). These examples highlight the platform’s unique strength in supporting exploratory analyses across granular, real-world variables, which is crucial for advancing equitable cancer care.

In addition to its analytical capacity, AI-HOPE-TP53 offers several design features that promote usability and reproducibility. The use of schema-guided natural language parsing and a retrieval-augmented generation (RAG) framework ensures consistent query interpretation and reduces the risk of LLM hallucination. Built-in audit trails, narrative result summaries, and multiple export options enhance transparency and support downstream communication in publications, presentations, and clinical workflows.

Importantly, AI-HOPE-TP53 has been tested by multidisciplinary users, confirming its accessibility to both computational and non-computational researchers. Its ability to deliver results in under 15 s without the need for programming or statistical software reflects its potential to democratize access to high-throughput data interrogation, particularly in community-based or resource-limited research settings.

While these findings demonstrate the broad utility of the platform, several limitations warrant consideration. First, because our analyses were derived exclusively from harmonized, publicly available datasets (e.g., TCGA, MSK, and GENIE), the findings may not fully represent real-world populations or treatment contexts. Certain high-risk groups, including rural, Indigenous, and uninsured patients, are underrepresented in these repositories, which limits the generalizability of some disparity-related findings. Therefore, the results from AI-HOPE-TP53 should be interpreted as hypothesis-generating signals that require confirmation in larger, ancestry-diverse, and prospectively collected cohorts. Second, the lack of statistical significance in several analyses reflects both limited sample sizes and the exploratory nature of this work. Future research will require larger, ancestry-diverse, and prospectively collected cohorts to confirm and extend these findings. Finally, the current version of AI-HOPE-TP53 focuses primarily on DNA-level alterations. In addition, while AI-HOPE-TP53 demonstrates technical feasibility and research utility, its current clinical applicability remains limited. The platform does not yet support direct integration of institutional patient-level datasets, electronic health records (EHRs), or multi-omic layers such as RNA-seq, proteomics, and epigenomics. As such, the immediate scope of AI-HOPE-TP53 is research-focused rather than clinical decision-making. Future development will prioritize these extensions to enhance the translational and clinical utility of this platform. Incorporation of transcriptomic, proteomic, and epigenomic layers, as well as compatibility with electronic health records (EHRs), would expand the platform’s clinical relevance and enhance multi-omic discovery.

In its current form, AI-HOPE-TP53 operates exclusively on harmonized, de-identified, and publicly available datasets, thereby ensuring full compliance with data privacy standards and avoiding the need for patient-level data sharing. While this design provides transparency, reproducibility, and accessibility across diverse research groups, it does not yet allow investigators to directly upload and analyze their own institutional datasets. Looking ahead, the modular architecture of AI-HOPE-TP53 offers a pathway for future development to support secure user-upload functionality. Such an extension would enable researchers to integrate their own clinical-genomic data within the platform’s framework while incorporating safeguards for privacy, Institutional Review Board (IRB) oversight, and data governance. This capability would not only expand the translational relevance of AI-HOPE-TP53 but also accelerate its adoption in real-world clinical and institutional research settings.

An important challenge of the current version of AI-HOPE-TP53 is the lack of built-in functionality for assessing statistical power and sample size adequacy prior to running analyses. While the platform reports subgroup counts alongside all results to help users gauge data availability, it does not yet provide formal power calculations or guidance on whether the available sample sizes are sufficient to detect meaningful effects. We recognize that incorporating such tools would substantially strengthen the robustness and interpretability of downstream analyses, particularly for disparity-aware subgroup comparisons, where sample sizes may be small. Because AI-HOPE-TP53 is designed with a modular architecture, future versions can readily integrate established power- and sample-size estimation modules. We view this as a priority for ongoing development to ensure that users can design and interpret analyses with greater statistical confidence.

Several findings from this study did not reach statistical significance, including the trend toward a higher prevalence of TP53 pathway alterations in H/L EOCRC patients than in NHW patients. While these results should not be interpreted as definitive, they are consistent with reported disparities in genomic burden among underrepresented populations and therefore serve as hypothesis-generating signals. The value of AI-HOPE-TP53 lies in its ability to detect and transparently report emerging patterns, which can guide future investigations with larger sample sizes, independent datasets, and prospective validation. Framing these outcomes as exploratory ensures appropriate caution while also underscoring the potential of the platform to highlight biologically and clinically relevant disparities for further study.

Another limitation of this study is that validation of AI-HOPE-TP53 was restricted to the replication of known genotype–phenotype associations within publicly available datasets. While this approach establishes feasibility and reproducibility, it does not provide independent or prospective confirmation of the platform’s findings. Future development will prioritize validation using institutional cohorts, prospective clinical datasets, and external resources such as GEO and ICGC, which will allow benchmarking of AI-HOPE-TP53 against real-world populations and treatment contexts that may be underrepresented in public repositories. In addition, integration of multi-omic layers—including RNA-seq, proteomic, and epigenomic data—will enable more comprehensive pathway interrogation. These steps are essential for ensuring the generalizability and translational relevance of AI-HOPE-TP53 and represent the next phase in the roadmap toward broader clinical and research applications. Future studies are critical to validate and expand the AI-HOPE-TP53 in real-world clinical settings. Specifically, testing the platform on institutional and prospectively collected clinical-genomic cohorts will allow for benchmarking against contemporary patient populations and treatment patterns. Integration of multi-omic data layers—including transcriptomics, proteomics, and epigenomics—will enhance biological interpretability and clinical precision. In addition, secure pipelines for interoperability with electronic health records (EHRs) will extend the relevance of this platform to clinical workflows. Finally, piloting AI-HOPE-TP53 across multi-institutional collaborations will provide opportunities to evaluate its scalability, usability, and equity-focused impact in diverse healthcare settings.

To ensure the long-term sustainability and relevance of AI-HOPE-TP53, we plan to establish a framework for regular dataset updates and LLM retraining. The modular design of the platform allows for seamless integration of new releases from public resources such as AACR GENIE, TCGA harmonizations, and MSK-IMPACT expansions, thereby keeping analyses aligned with the most current data available. In parallel, the underlying large language model can be periodically fine-tuned with updated biomedical corpora to incorporate emerging scientific knowledge and evolving clinical standards. By implementing a scheduled update cycle, we aim to maintain the accuracy, reproducibility, and translational utility of AI-HOPE-TP53, ensuring that it continues to serve as a reliable and up-to-date resource for pathway-centric and disparity-aware EOCRC research.

The emergence of AI-HOPE-TP53 reflects a broader evolution in cancer research, where artificial intelligence (AI) is harnessed to bridge the gap between high-throughput data and clinically meaningful insights. Across domains, AI is transforming how researchers interpret complex biomedical signals—from radiomics in breast cancer imaging [41] to drug discovery pipelines leveraging deep learning and knowledge graph models [42,43,44]. In this context, AI-HOPE-TP53 complements existing AI approaches by uniquely focusing on pathway-specific and disparity-aware interrogation of CRC. The platform’s ability to model TP53-related molecular dynamics within population subgroups, such as early onset and Hispanic/Latino patients, aligns with recent multi-omics studies that have uncovered enrichment of MYC, WNT, and TP53 pathway alterations in these underrepresented cohorts [45,46]. Additionally, the usability of AI-HOPE-TP53 mirrors trends in patient-centered conversational agents, which are being increasingly integrated into cancer education and care delivery to enhance engagement and accessibility [47]. As multimodal technologies gain traction in clinical practice [48], tools like AI-HOPE-TP53 provide a blueprint for real-time, natural language–driven integration of clinical, genomic, and sociodemographic data, with immediate implications for both discovery and translational equity. As we continue to unravel the interplay of signaling cascades, such as β-catenin and RAF-MEK [49,50], and as spatial biology becomes more prominent in precision oncology [51], conversational AI systems like AI-HOPE-TP53 will be well-positioned to incorporate new data layers and guide pathway-specific insights across the cancer research continuum. By embedding equity-aware analytics and open-access usability, AI-HOPE-TP53 not only accelerates hypothesis generation but also reinforces the growing imperative to democratize AI in cancer research.

## 5. Conclusions

AI-HOPE-TP53 introduces a freely accessible, natural language–driven platform for pathway-centric, disparity-aware exploration of colorectal cancer genomics. Through multiple use cases, the platform demonstrated its capacity to replicate known associations and uncover potential differences across age, ancestry, tumor subsites, treatment exposure, and sex. While several analyses revealed statistically significant findings, the majority of results are exploratory and should be interpreted as hypothesis-generating signals rather than confirmatory evidence.

The primary value of AI-HOPE-TP53 lies in its ability to democratize access to clinical-genomic interrogation, accelerate disparity-aware hypothesis generation, and support equity-focused precision oncology research. Future studies using larger, ancestry-diverse, and prospectively collected cohorts, along with the integration of multi-omic and real-world clinical data, will be critical for validaing and extending these findings. By enabling rapid, user-friendly, and reproducible analyses, AI-HOPE-TP53 provides a foundation for ongoing development toward clinically relevant and translational applications, while currently serving as a robust exploratory resource for the cancer research community.

## Figures and Tables

**Figure 1 cancers-17-02865-f001:**
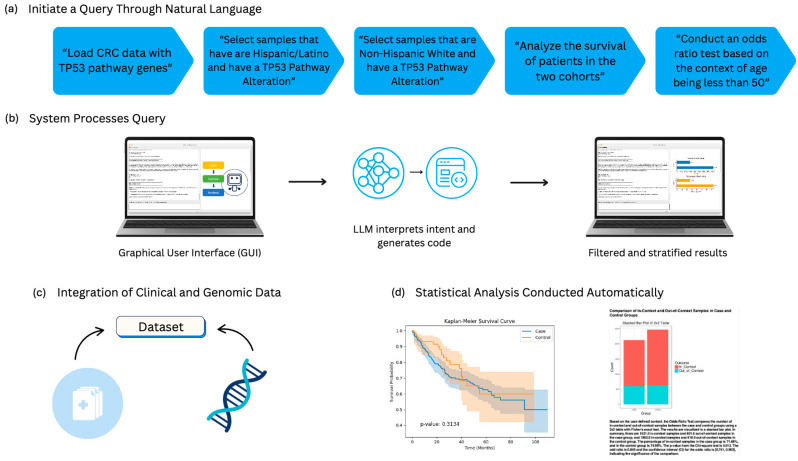
Schematic representation of AI-HOPE-TP53 query and analysis pipeline. This figure outlines the core operational stages of AI-HOPE-TP53, a conversational AI agent designed to analyze TP53-related genomic events in colorectal cancer (CRC) through integrative, automated workflows. (**a**) The process begins with a series of sequential natural language inputs, where users specify analysis parameters, such as cohort definitions (e.g., TP53-altered Hispanic or Latino vs. non-Hispanic White patients), survival outcome comparisons, or subgroup-specific odds ratio testing for early onset cases. (**b**) These inputs are parsed by a graphical user interface (GUI) that channels them through a backend engine, where an LLM translates the user’s intent into code-based instructions and dynamically builds filtered clinical-genomic subsets. (**c**) AI-HOPE-TP53 connects to curated datasets, including cBioPortal, and extracts relevant molecular and phenotypic information centered on TP53 and its associated regulatory genes (e.g., CDKN1A, GADD45A, MDM2, BAX, and ATM). (**d**) Once executed, the platform returns fully stratified results and automated statistical outputs, including Kaplan-Meier survival curves, contingency tables, and effect size visualizations.

**Figure 2 cancers-17-02865-f002:**
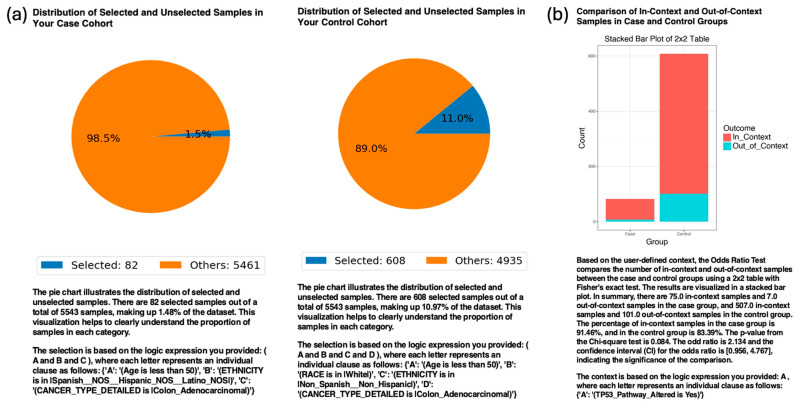
AI-HOPE-TP53 analysis of TP53 pathway alterations in early onset colon adenocarcinoma among Hispanic/Latino (H/L) and Non-Hispanic White (NHW) patients. (**a**) Pie charts illustrate the distribution of selected samples following natural language–driven filtering of the dataset. The case cohort included 82 patients with early onset colorectal cancer (EOCRC) patients under age 50 years of H/L ethnicity and a diagnosis of colon adenocarcinoma, comprising 1.5% of the dataset. The control cohort included 608 patients under age 50 years who were NHW and diagnosed with colon adenocarcinoma, representing 11% of the dataset. (**b**) A 2 × 2 odds ratio analysis was used to evaluate the frequency of TP53 pathway alterations between the two groups. The stacked bar plot shows the proportion of samples with and without TP53 pathway alterations. Alterations were present in 91.46% of H/L cases and 83.39% of NHW controls. The calculated odds ratio was 2.13 (95% CI: 0.956–4.767), with a *p*-value of 0.084, indicating a non-significant but suggestive trend toward a greater TP53 alteration prevalence in the H/L population.

**Figure 3 cancers-17-02865-f003:**
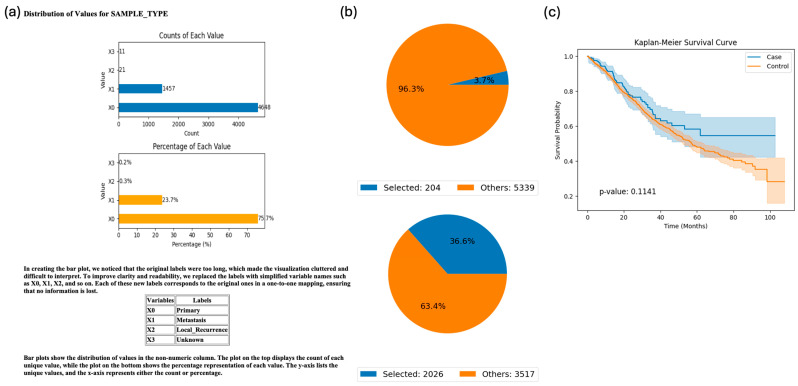
AI-HOPE-TP53 analysis of TP53-mutant primary colorectal cancer samples by ethnicity (Hispanic/Latino-H/L vs. Non-Hispanic White-NHW). This figure demonstrates the use of AI-HOPE-TP53 to assess survival outcomes in patients with TP53-mutated colorectal cancer (CRC), stratified by ethnicity and limited to primary-tumor samples. (**a**) Bar plots summarizing the distribution of sample types across the dataset. The majority of samples are classified as primary tumors (X0), accounting for 75.7% of cases (*n* = 4648), followed by metastatic samples (X1, 23.7%, *n* = 1457). Labels were condensed for clarity, with X0 representing primary, X1 representing metastasis, and less common categories (e.g., local recurrence, unknown) grouped into X2 and X3. (**b**) Pie charts visualize the cohort selection process after applying natural language query filters. The case cohort (H/L patients with TP53-mutated primary tumors) includes 204 samples (3.7% of the dataset), while the control cohort (NHW patients with TP53-mutated primary tumors) includes 2026 samples (36.6%). These charts provide context for subgroup sizes and selection criteria. (**c**) Kaplan-Meier survival curves comparing the overall survival between the two groups. While H/L patients with TP53-mutated primary CRC show a trend toward improved survival, the difference was not statistically significant (*p* = 0.1141). Shaded regions represent 95% confidence intervals.

**Figure 4 cancers-17-02865-f004:**
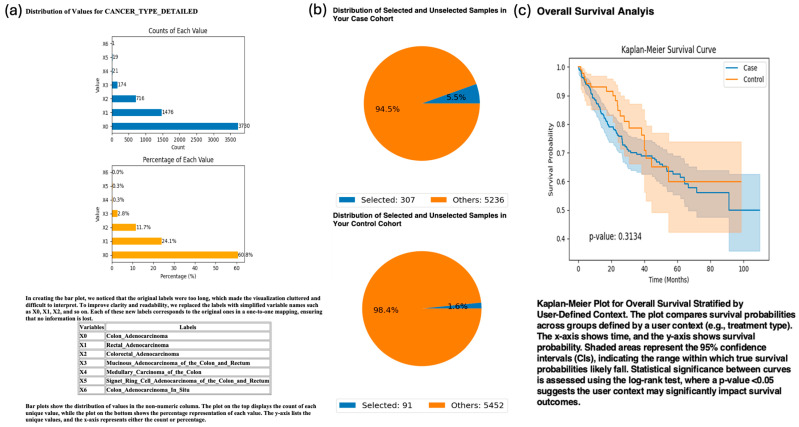
AI-HOPE-TP53 Analysis of ATM-Mutated Colorectal Cancer (CRC) Stratified by Tumor Subsite: Colon vs. Rectal Adenocarcinoma. This figure demonstrates the application of AI-HOPE-TP53 to evaluate survival outcomes among CRC patients harboring ATM mutations, stratified by tumor origin—colon adenocarcinoma (case) versus rectal adenocarcinoma (control). (**a**) Bar plots display the distribution of detailed cancer subtypes across the dataset. Colon adenocarcinoma (X0) is the most prevalent subtype, comprising 60.6% of samples (*n* = 3370), followed by rectal adenocarcinoma (X1, 24.1%, *n* = 1341). Labels were simplified to improve clarity (e.g., X0 = Colon_Adenocarcinoma, X1 = Rectal_Adenocarcinoma). The top panel shows the sample counts, and the lower panel shows their proportional representation. (**b**) Pie charts highlight the selection of ATM-mutated samples for each group. The case cohort (colon adenocarcinoma with ATM mutations) includes 307 samples (5.5% of the dataset), and the control cohort (rectal adenocarcinoma with ATM mutations) includes 91 samples (1.6%). These distributions provide a visual context for the relative representation of ATM-mutated cases across tumor subsites. (**c**) Kaplan-Meier survival curves comparing overall survival between ATM-mutated colon and rectal cancer cohorts. The case group (colon adenocarcinoma) showed a trend toward poorer survival than the control group, but the difference was not statistically significant (*p* = 0.3134). Confidence intervals are shown as shaded bands.

**Figure 5 cancers-17-02865-f005:**
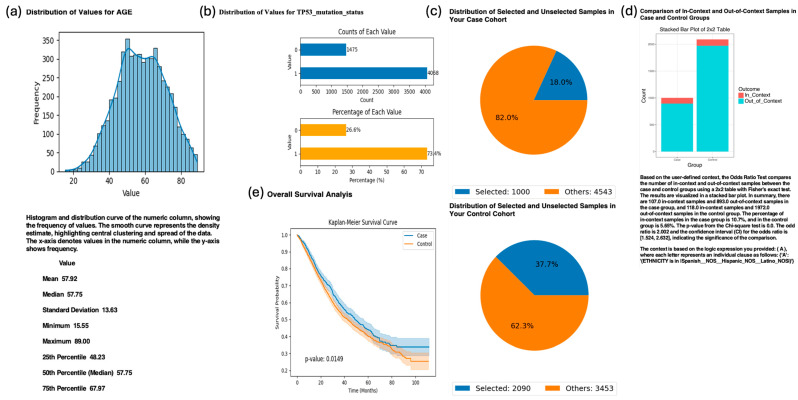
AI-HOPE-TP53 Analysis of TP53-Mutated Colorectal Cancer (CRC) Patients Treated with FOLFOX Chemotherapy, Stratified by Age and Evaluated for Hispanic/Latino (H/L) Representation. This figure demonstrates the use of AI-HOPE-TP53 to assess survival outcomes and ethnic enrichment among TP53-mutated CRC patients treated with the FOLFOX chemotherapy regimen (Fluorouracil, Leucovorin, and Oxaliplatin), comparing early onset (<50 years) and late-onset (>50 years) cases. (**a**) A histogram depicting the age distribution across the full dataset, with a mean of 57.92 years and a median of 57.75. This distribution contextualizes the cutoff for classifying early onset CRC. (**b**) Bar plots summarizing the dataset-wide distribution of TP53 mutation status. TP53-mutated samples (X0) represent 26.5% of the cohort, while wild-type samples (X1) make up the remaining 73.5%. The top panel shows the absolute counts, while the bottom panel shows the proportional values. (**c**) Pie charts illustrate the cohort selection process. The case cohort includes 1000 early-onset CRC patients (age < 50) with TP53 mutations treated with FOLFOX (18.0% of the dataset), while the control group consists of 2090 late-onset CRC patients (age > 50) with the same molecular and treatment profile (37.7%). (**d**) A 2 × 2 odds ratio analysis evaluated the enrichment of H/L patients within the two cohorts. Among early-onset TP53-mutated patients treated with FOLFOX, 10.7% were H/L compared to 5.65% in the late-onset group. The calculated odds ratio was 2.002 (95% CI: 1.524–2.632; *p* < 0.0001), indicating statistically significant enrichment of H/L individuals in the early-onset group. (**e**) Kaplan-Meier survival analysis compares overall survival between the case and control groups. Patients with early onset disease demonstrated significantly improved survival outcomes (*p* = 0.0149), with clear separation between the curves and non-overlapping confidence intervals.

## Data Availability

Data used in this study are publicly available at cbioportal.org. The AI-HOPE-TP53 software, along with demonstration data and documentation, is available at https://github.com/Velazquez-Villarreal-Lab/AI-TP53 (accessed on 2 August 2025).

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
