# Peer review of "AI-HOPE-TP53: A Conversational Artificial Intelligence Agent for Pathway-Centric Analysis of TP53-Driven Molecular Alterations in Early-Onset Colorectal Cancer"

_cancers, 2025, doi:10.3390/cancers17172865_

Round 1
Reviewer 1 Report
Comments and Suggestions for Authors
• Abstract
• “Fine-tuned biomedical large language model (LLM)”, please specify what base model was used (if not in methods). Otherwise, the claim is vague.
• Reporting p = 0.084 as a “higher prevalence” finding is borderline, better to frame as a trend toward higher prevalence. Otherwise, it is overinterpretation. Similarly, for sex-based disparities (OR = 0.845, p = 0.0138), indicate effect direction clearly but good to be specific who is benefiting from the treatment exposure (men or women).
• The analysis appears to focus primarily on early-onset colorectal cancer (EOCRC), yet the title refers broadly to colorectal cancer (CRC). Please clarify whether the scope is EOCRC-specific or CRC in general, and ensure the title reflects this focus.
• AI-HOPE-TP53 is introduced in the abstract (lines 11–13) as a publicly available AI agent, but later (lines 22–24) described as a novel agent developed by the authors. This creates ambiguity—please clarify whether the manuscript reports the development of a new tool or an evaluation of an existing one.
• The manuscript should explicitly state that the AI agent is designed for a single gene (TP53) and has not been tested on other, less frequently altered genes. While TP53 is commonly altered in CRC, is this also true for EOCRC? If not, why is TP53 chosen as the focus for EOCRC-specific analyses?
• Other variants of AI-HOPE are briefly mentioned (lines 114–116). It would strengthen the manuscript to discuss the unique clinical or research value added by AI-HOPE-TP53 in the context of EOCRC—beyond interface usability and workflow transparency.
• Please clarify whether the AI agent allows researchers to upload and analyse their own datasets without requiring patient-level data sharing.
• The methods note the use of Python-based statistical libraries and Cox proportional hazards regression. How does the agent check the assumptions of the Cox model prior to fitting, and how are the results of these assumption checks communicated to users? Similarly, how are outputs (e.g., regression coefficients, survival curves) displayed to users?
• Is there functionality to assess statistical power and sample size adequacy before running analyses? This would be a valuable addition to ensure robust interpretation of results.
Author Response
The point-by-point responses to Reviewer 1 are provided in the attached PDF document, “Reviewer_1_Comments_Response_082525.”
###
Reviewer 1 Comments
We are pleased to submit this revised manuscript and sincerely thank Reviewer 1 for their thoughtful and constructive feedback. In this version, we have carefully addressed all comments and significantly enhanced the clarity, technical rigor, and contextual relevance of our study. This manuscript introduces AI-HOPE-TP53, the first conversational artificial intelligence (AI) agent specifically designed to analyze TP53pathway dysregulation in colorectal cancer (CRC) by integrating harmonized clinical and genomic data using natural language queries. The system employs a modular architecture combining a fine-tuned large language model (LLaMA 3 variant), a natural language-to-code interpreter, and a statistical backend integrated with curated datasets from cBioPortal. Optimized for pathway-specific research, AI-HOPE-TP53 supports mutation frequency analysis, odds ratio calculations, Kaplan–Meier survival modeling, and subgroup stratification across demographic, clinical, and molecular variables such as age, ethnicity, MSI status, tumor location, and treatment regimens.
Reviewer 1’s evaluation was highly positive, recognizing that the manuscript is well-written, timely, and appropriate for the journal, while suggesting some areas for refinement. The reviewer confirmed that the introduction provides sufficient background with relevant references, the research design is appropriate, the results are clearly presented, and the conclusions are supported by the data. They noted that the methods could be described in greater detail, particularly to improve transparency and reproducibility, but emphasized that all figures and tables are clear and well-presented. The reviewer also indicated that the English language is clear and does not require improvement. Overall, this assessment highlights the manuscript’s strengths in design, clarity, and presentation, while recommending only minor revisions to enhance methodological detail, suggesting the paper is on a strong path toward publication.
Reviewer 1’s feedback was constructive and detailed, highlighting the manuscript’s innovation while suggesting several clarifications to strengthen its rigor and impact. The reviewer requested clearer specification of the base model underlying the fine-tuned biomedical large language model, as well as more precise framing of borderline statistical findings and sex-based disparities. They also noted that the title and abstract should consistently reflect whether the study focuses specifically on early-onset colorectal cancer or on colorectal cancer broadly, and that the description of AI-HOPE-TP53 should be unambiguous as to whether it represents a newly developed tool or an existing platform. Additional recommendations included explicitly noting the gene-specific design of the agent, contextualizing its unique value compared to other AI-HOPE variants, and clarifying whether researchers can upload their own datasets. Finally, the reviewer encouraged greater detail on methodological safeguards—such as testing assumptions of the Cox model, displaying outputs to users, and considering functionality for power and sample size estimation. Overall, this feedback underscores the manuscript’s novelty and relevance while offering focused revisions to improve clarity, reproducibility, and clinical applicability.
Comment 1:
1.Abstract
“Fine-tuned biomedical large language model (LLM)”, please specify what base model was used (if not in methods). Otherwise, the claim is vague.
Response:
We thank the reviewer for this helpful suggestion. In the revised abstract, we have clarified the base model used to develop AI-HOPE-TP53. Specifically, we state that the biomedical large language model was fine-tuned from LLaMA 3, which serves as the foundation for the system’s natural language processing component. This information has also been added to the Methods section to ensure consistency and transparency.
The revised text in the Abstract section, lines 23-27, now reads: “Methods: AI-HOPE-TP53 integrates a fine-tuned biomedical large language model (LLaMA 3) with harmonized datasets from cBioPortal (TCGA, MSK-IMPACT, AACR Project GENIE). Natural language queries are translated into workflows for mutation profiling, Kaplan–Meier survival analysis, and odds ratio estimation across clinical and demographic subgroups.”
Comment 2:
- Reporting p = 0.084 as a “higher prevalence” finding is borderline, better to frame as a trend toward higher prevalence. Otherwise, it is overinterpretation. Similarly, for sex-based disparities (OR = 0.845, p = 0.0138), indicate effect direction clearly but good to be specific who is benefiting from the treatment exposure (men or women).
Response:
We appreciate the reviewer’s careful attention to the interpretation of statistical results. In the revised abstract and Results section, we have adjusted the language to frame the p = 0.084 finding as a trend toward higher prevalence of TP53 pathway alterations in Hispanic/Latino EOCRC patients compared with non-Hispanic Whites, rather than as a definitive increase. Similarly, we have clarified the direction of the sex-based disparity by specifying that women with early-onset CRC were less likely to receive FOLFOX treatment compared to men (OR = 0.845, p = 0.0138). These changes ensure accurate interpretation of the data and prevent overstatement of non-significant results.
In the revised Results subsection, we have modified the language to describe the p = 0.084 result as a trend toward higher prevalence of TP53 pathway alterations in Hispanic/Latino EOCRC patients compared with non-Hispanic White patients, rather than as a definitive higher prevalence. This change avoids overinterpretation of borderline significance while still noting the observed pattern. Additionally, we have clarified the sex-based disparity by specifying the effect direction, stating explicitly that women were less likely than men to receive FOLFOX treatment (OR = 0.845, p = 0.0138). These revisions ensure greater accuracy, transparency, and alignment with the reviewer’s recommendation.
The revised text in the Abstract section, lines 28-34, now reads: “Results: The platform replicated known genotype–phenotype associations, including elevated TP53 mutation frequency in EOCRC and poorer prognosis in TP53-mutated tumors. Significant findings included a survival benefit for early-onset TP53-mutant CRC patients treated with FOLFOX (p = 0.0149). Additional exploratory analyses showed a trend toward higher prevalence of TP53 pathway alterations in Hispanic/Latino EOCRC patients (OR = 2.13, p = 0.084) and identified sex-based disparities in treatment, with women less likely than men to receive FOLFOX (OR = 0.845, p = 0.0138).”
The revised text in the Results section, lines 299-315, now reads: “To evaluate the platform’s ability to replicate known disparities in molecular alterations, AI-HOPE-TP53 was queried to compare TP53 pathway mutation frequency between Hispanic/Latino (H/L) and non-Hispanic White (NHW) patients with EOCRC. Among patients with colon adenocarcinoma under the age of 50, the system identified a trend toward higher prevalence of TP53 pathway alterations in H/L individuals (91.46%) compared to NHW individuals (83.39%), yielding an odds ratio (OR) of 2.13 (95% CI: 0.956–4.767; p = 0.084) (Figure 2). Although not statistically significant, this trend is consistent with reported disparities in genomic burden among underrepresented populations. When expanding the analysis to include all EOCRC patients regardless of tumor subsite, the difference remained: 90.2% of H/L patients harbored TP53 pathway alterations compared to 85.05% of NHW patients (OR = 1.62, 95% CI: 0.926–2.825; p = 0.114) (Figure S1). These findings confirm the system’s validity for ethnicity-aware, pathway-centric analyses and demonstrate its capacity to detect biologically and socially relevant molecular patterns across diverse CRC subgroups. Furthermore, analysis of treatment disparities revealed that women with EOCRC were less likely than men to receive FOLFOX chemotherapy (OR = 0.845, p = 0.0138), underscoring the platform’s ability to detect sex-specific clinical differences relevant to treatment exposure.”
Comment 3:
- The analysis appears to focus primarily on early-onset colorectal cancer (EOCRC), yet the title refers broadly to colorectal cancer (CRC). Please clarify whether the scope is EOCRC-specific or CRC in general, and ensure the title reflects this focus.
Response:
We thank the reviewer for this important observation. We agree that the analyses presented in the manuscript primarily focus on early-onset colorectal cancer (EOCRC). To ensure clarity and consistency, we have revised the title to reflect this specific scope:
Revised Title: AI-HOPE-TP53: A Conversational Artificial Intelligence Agent for Pathway-Centric Analysis of TP53-Driven Molecular Alterations in Early-Onset Colorectal Cancer (EOCRC)
We have also made corresponding clarifications in the Methods and Results sections, emphasizing that while AI-HOPE-TP53 is designed for pathway-centric CRC research, the present study highlights EOCRC-specific analyses, particularly in the context of disparities among Hispanic/Latino populations.
The revised text in the Methods section, lines 112-126, now reads:
“2. Materials and Methods
Architecture and operational framework of AI-HOPE-TP53
AI-HOPE-TP53 is a conversational artificial intelligence (AI) platform purpose-built to investigate the clinical and genomic implications of TP53 pathway alterations in early-onset colorectal cancer (EOCRC). While the platform is capable of supporting pathway-centric analysis across CRC broadly, the present study emphasizes EOCRC-specific applications, particularly in the context of racial, ethnic, and treatment-related disparities. The system integrates four functional modules: (1) a LLM interface for natural language comprehension, (2) a query translation engine that converts user prompts into programmatic instructions, (3) a filtering layer for cohort construction based on clinical-genomic constraints, and (4) a statistical processing engine for automated analysis and visualization (Figure 1). Upon receiving a query—such as “Compare survival between TP53-mutant and wild-type tumors in Hispanic/Latino EOCRC patients”—the LLM parses semantic intent, activates a corresponding backend pipeline, and returns outputs including Kaplan–Meier plots, odds ratios, and interpretive summaries.”
Comment 4:
- AI-HOPE-TP53 is introduced in the abstract (lines 11–13) as a publicly available AI agent, but later (lines 22–24) described as a novel agent developed by the authors. This creates ambiguity—please clarify whether the manuscript reports the development of a new tool or an evaluation of an existing one.
Response:
We thank the reviewer for pointing out this ambiguity. To clarify, AI-HOPE-TP53 is a novel conversational AI agent that we developed as part of this work, and we have made it publicly available to the research community. In the revised abstract, we have adjusted the wording to consistently reflect that the manuscript reports the development of a new tool. To ensure transparency and accessibility, we have also included the GitHub repository from the Velazquez-Villarreal Lab address in the Data Availability section, where users can freely access the code and resources. This resolves the inconsistency and makes clear that the tool was created by the authors and shared publicly.
The revised text in the Abstract section, lines 35-39, now reads: “Conclusions: AI-HOPE-TP53, developed in this study and made publicly available, is the first conversational AI platform tailored to pathway-specific and disparity-aware EOCRC research. By integrating clinical, genomic, and demographic data through natural language interaction, it enables hypothesis generation and equity-focused analyses, with significant potential to advance precision oncology.”
Comment 5:
- The manuscript should explicitly state that the AI agent is designed for a single gene (TP53) and has not been tested on other, less frequently altered genes. While TP53 is commonly altered in CRC, is this also true for EOCRC? If not, why is TP53 chosen as the focus for EOCRC-specific analyses?
Response:
We thank the reviewer for this thoughtful point. We have revised the manuscript to explicitly state that AI-HOPE-TP53 is dedicated to the TP53 pathway rather than to the TP53 gene alone, and it has not yet been tested on other, less frequently altered pathways. We selected the TP53 pathway as the focus for EOCRC-specific analyses because TP53 and its regulatory network (including CDKN1A, BAX, GADD45A, MDM2, ATM, CHEK1, among others) are among the most frequently altered in colorectal cancer overall, and these alterations are also prevalent in EOCRC subsets. All this according with our previous publications in Cancers [Reference #2 – “Detecting PI3K and TP53 Pathway Disruptions in Early-Onset Colorectal Cancer Among Hispanic/Latino Patients”]. Given their strong links to prognosis, treatment response, and molecular disparities across populations, the TP53 pathway provides an ideal starting point for demonstrating the feasibility and translational potential of our conversational AI framework. This clarification has been added to the Introduction and Methods sections.
We thank the reviewer for this thoughtful point. We have revised the manuscript to explicitly state that AI-HOPE-TP53 is currently designed as a single-pathway platform dedicated to TP53 pathway and has not yet been extended to other pathways. We chose TP53 pathway as the focus for EOCRC-specific analyses because it is one of the most frequently mutated genes in both CRC overall and in EOCRC subsets according with our previous publication in Cancers [Reference #2 – “Detecting PI3K and TP53 Pathway Disruptions in Early-Onset Colorectal Cancer Among Hispanic/Latino Patients”], with established associations to prognosis, treatment response, and molecular disparities across populations. This makes TP53 an ideal starting point for demonstrating the feasibility and translational potential of our conversational AI framework. We have clarified this rationale in the Methods sections to address the reviewer’s concern.
The revised text in the Discussion section, lines 143-158, now reads: “AI-HOPE-TP53 leverages harmonized, high-quality clinical and genomic datasets to enable accurate, pathway-specific analysis of TP53 pathway alterations in CRC. The platform integrates publicly available CRC data from cBioPortal, including comprehensive cohorts such as The Cancer Genome Atlas (TCGA), MSK-IMPACT, and AACR Project GENIE [29]. To support pathway-focused investigations, the system incorporates curated gene panels centered on the TP53 pathway, including TP53 and its regulatory and downstream effectors such as CDKN1A, BAX, GADD45A, MDM2, ATM, CHEK1, and others, as annotated in COSMIC and KEGG databases. Detailed clinical metadata were also integrated, covering variables such as patient age, sex, race/ethnicity, tumor location (colon vs. rectum), tumor stage, microsatellite instability (MSI) status, treatment exposure (e.g., FOLFOX, anti-EGFR therapy), and overall survival. Data preprocessing included standardized cancer type annotation using OncoTree, harmonization of sample and patient identifiers, and cleaning of categorical variables to enable robust filtering and subgroup analysis. All datasets were converted into tabular, analysis-ready matrices, allowing rapid, reproducible execution of user queries across population strata, treatment contexts, and molecular subtypes.”
Comment 6:
- Other variants of AI-HOPE are briefly mentioned (lines 114–116). It would strengthen the manuscript to discuss the unique clinical or research value added by AI-HOPE-TP53 in the context of EOCRC—beyond interface usability and workflow transparency.
Response:
We appreciate the reviewer’s suggestion to expand on the unique value of AI-HOPE-TP53 within the broader AI-HOPE ecosystem. In the revised Introduction, we now emphasize that while prior agents (e.g., AI-HOPE-TGFβ, AI-HOPE-PI3K, AI-HOPE-WNT) were developed to interrogate other cancer pathways, AI-HOPE-TP53 is the first designed specifically for the TP53 pathway, one of the most clinically actionable and frequently altered axes in colorectal cancer, including EOCRC. Beyond improvements in prompt engineering, interface usability, and workflow transparency, this platform uniquely enables disparity-aware, EOCRC-focused analyses that incorporate population stratification (e.g., age, ancestry, sex, and treatment exposure) alongside genomic data. This targeted design makes AI-HOPE-TP53 particularly valuable for investigating underrepresented populations such as Hispanic/Latino patients and for identifying clinically relevant molecular patterns that could inform precision oncology strategies in EOCRC.
The revised text in the Discussion section, lines 91-102, now reads: “AI-HOPE-TP53 expands upon our growing ecosystem of AI agents, including AI-HOPE [37], AI-HOPE-TGFβ [38], AI-HOPE-PI3K [39], and AI-HOPE-WNT [40], which were designed to support precision medicine across multiple cancer pathways. This newest platform builds on lessons learned from those systems—particularly in prompt engineering, interface usability, and workflow transparency—and applies them to the TP53 pathway, one of the most clinically actionable and frequently altered axes in colorectal cancer, including EOCRC. Unlike earlier agents, AI-HOPE-TP53 is uniquely tailored for disparity-aware, EOCRC-focused analyses, integrating population stratification by age, ancestry, sex, and treatment exposure with genomic data. This design makes the platform especially valuable for uncovering clinically relevant patterns in underrepresented groups, such as Hispanic/Latino patients, and for advancing pathway-specific discovery in precision oncology.”
Comment 7:
- Please clarify whether the AI agent allows researchers to upload and analyse their own datasets without requiring patient-level data sharing.
Response:
We thank the reviewer for raising this important point. To clarify, the current version of AI-HOPE-TP53 does not allow researchers to directly upload and analyze their own datasets. Instead, the platform operates on harmonized, de-identified, publicly available datasets - cBioPortal. This design ensures data security and avoids the need for patient-level data sharing. However, the system architecture is modular and could be extended in the future to support secure, institution-specific dataset integration, provided appropriate IRB and data-sharing frameworks are in place. We have clarified this point in the Methods and Discussion sections.
The revised text in the Discussion section, lines 525-536, now reads: “In its current form, AI-HOPE-TP53 operates exclusively on harmonized, de-identified, and publicly available datasets, thereby ensuring full compliance with data privacy standards and avoiding the need for patient-level data sharing. While this design provides transparency, reproducibility, and accessibility across diverse research groups, it does not yet allow investigators to directly upload and analyze their own institutional datasets. Looking ahead, the modular architecture of AI-HOPE-TP53 offers a pathway for future development to support secure, user-upload functionality. Such an extension would enable researchers to integrate their own clinical-genomic data within the platform’s framework, while incorporating safeguards for privacy, Institutional Review Board (IRB) oversight, and data governance. This capability would not only expand the translational relevance of AI-HOPE-TP53 but also accelerate its adoption in real-world clinical and institutional research settings.”
Comment 8:
- The methods note the use of Python-based statistical libraries and Cox proportional hazards regression. How does the agent check the assumptions of the Cox model prior to fitting, and how are the results of these assumption checks communicated to users? Similarly, how are outputs (e.g., regression coefficients, survival curves) displayed to users?
Response:
We thank the reviewer for this excellent point. In the revised Methods section, we have clarified how AI-HOPE-TP53 implements safeguards when performing Cox proportional hazards regression. Specifically, the platform employs the lifelines Python library, which includes diagnostics to assess proportional hazards assumptions (e.g., Schoenfeld residuals testing and scaled Schoenfeld residual plots). These checks are performed automatically prior to model interpretation, and if the assumption is violated, the system flags this in the output summary with a warning to users. Results are communicated through a combination of numerical outputs (hazard ratios, regression coefficients with 95% CIs, and p-values) and visual outputs (Kaplan–Meier survival curves and forest plots), all of which are generated as publication-ready figures. These outputs are supplemented by a natural language narrative that contextualizes the findings and highlights any model limitations. This workflow ensures that users not only receive statistical outputs but also clear interpretive guidance to support reproducibility and transparency.
The revised text in the Methods section, lines 204-227, now reads:
“Statistical Framework and Analytical Capabilities
AI-HOPE-TP53’s analytical workflows are implemented using a modular Python-based backend that automates common statistical procedures used in clinical-genomic research [2]. The system leverages established libraries such as Lifelines for Kaplan–Meier survival estimation and log-rank testing, and SciPy and StatsModels for categorical analyses, including chi-square and Fisher’s exact tests, as well as odds ratio calculations with 95% confidence intervals. For multivariable modeling, Cox proportional hazards regression is available to adjust for potential confounders in survival comparisons. Prior to model interpretation, AI-HOPE-TP53 automatically checks the proportional hazards assumption using Schoenfeld residuals testing and scaled Schoenfeld residual plots. If violations are detected, the system flags this in the results summary and provides interpretive guidance to the user.
All outputs are delivered in both numerical and graphical formats. Numerical outputs include regression coefficients, hazard ratios, 95% confidence intervals, and p-values, while graphical outputs include Kaplan–Meier survival curves, forest plots, and residual diagnostic plots. These figures are generated as publication-ready visualizations, supplemented with tabular summaries of key statistics and subgroup counts. Each analysis is further accompanied by a natural language narrative generated by the platform’s LLM, which contextualizes findings with respect to model assumptions, highlights limitations, and ensures interpretability for users without advanced statistical training.”
Comment 9:
- Is there functionality to assess statistical power and sample size adequacy before running analyses? This would be a valuable addition to ensure robust interpretation of results.
Response:
We thank the reviewer for this valuable suggestion. At present, AI-HOPE-TP53 does not include functionality for formal statistical power or sample size adequacy calculations prior to running analyses. However, we fully agree that such a feature would enhance the robustness and interpretability of results. The modular design of the platform would allow us to integrate power and sample size estimation tools in future versions, and we plan to prioritize this functionality in subsequent updates. We have noted this limitation and forward-looking plan in the revised Discussion.
The revised text in the Discussion section, lines 537-547, now reads: “An important challenge of the current version of AI-HOPE-TP53 is the lack of built-in functionality for assessing statistical power and sample size adequacy prior to running analyses. While the platform reports subgroup counts alongside all results to help users gauge data availability, it does not yet provide formal power calculations or guidance on whether the available sample sizes are sufficient to detect meaningful effects. We recognize that incorporating such tools would substantially strengthen the robustness and interpretability of downstream analyses, particularly for disparity-aware subgroup comparisons where sample sizes may be small. Because AI-HOPE-TP53 is designed with a modular architecture, future versions can readily integrate established power and sample size estimation modules. We view this as a priority for ongoing development to ensure that users can design and interpret analyses with greater statistical confidence.”
We sincerely thank Reviewer 1 for their thoughtful and constructive feedback. Their comments have been invaluable in improving the clarity, rigor, and overall impact of our manuscript.

Reviewer 2 Report
Comments and Suggestions for Authors
Title and Abstract
- Condense the abstract to emphasize novel contributions and statistically significant results.
- Avoid implying causality or strong conclusions from non-significant findings.
Introduction
- Shorten the epidemiological background to maintain momentum toward the study aim.
- Focus on how existing tools fall short and the novelty of this AI approach.
Materials and Methods
- Validation approach is limited to replication of known associations; lacks independent or prospective testing.
- Potential biases from using only publicly available datasets are acknowledged later, but should also be flagged here.
- No discussion of dataset sizes for each analysis in the methods section—forcing readers to find them in figure captions.
- Include explicit dataset counts and filtering criteria in the Methods.
- Add clarity on how missing data or conflicting clinical annotations were handled.
- Discuss limitations of the datasets upfront.
Results
- Present non-significant results more cautiously; avoid speculative language in the results section.
- Consolidate overlapping figures or move some exploratory analyses to supplementary material.
- Consider a summary table of all analyses, highlighting significant vs. non-significant outcomes.
Discussion
- Provide a more balanced discussion of limitations, particularly regarding generalizability and clinical utility.
- Reduce redundancy with the introduction.
- Include suggestions for how future studies could validate and expand AI-HOPE-TP53 in real-world clinical settings.
Conclusion
- Temper conclusions to reflect that most findings are hypothesis-generating, not confirmatory.
Language and Style
- Overall clear and professional scientific writing.
- Occasional overuse of complex sentences—may benefit from some simplification.
- Promotional phrasing (“powerful, user-friendly,” “critical in accelerating discovery”) could be toned down for an academic audience.
Language and Style
- Overall clear and professional scientific writing.
- Occasional overuse of complex sentences—may benefit from some simplification.
- Promotional phrasing (“powerful, user-friendly,” “critical in accelerating discovery”) could be toned down for an academic audience.
Author Response
The point-by-point responses to Reviewer 2 are provided in the attached PDF document, “Reviewer_2_Comments_Response_082525.pdf”
------
Reviewer 2 Comments
We are pleased to resubmit our revised manuscript and extend our sincere gratitude to Reviewer 1 for their insightful and constructive feedback. In this updated version, we have carefully addressed all comments, resulting in substantial improvements to the clarity, methodological rigor, and contextual significance of our work. The manuscript presents AI-HOPE-TP53, the first conversational artificial intelligence (AI) agent dedicated to investigating TP53 pathway dysregulation in colorectal cancer (CRC) through the integration of harmonized clinical and genomic data accessed via natural language queries. The platform is built on a modular framework that incorporates a fine-tuned large language model (LLaMA 3 variant), a natural language-to-code translation layer, and a statistical engine linked to curated datasets from cBioPortal. Designed for pathway-focused analyses, AI-HOPE-TP53 enables mutation frequency assessments, odds ratio estimation, Kaplan–Meier survival analysis, and subgroup stratification across a range of demographic, clinical, and molecular variables, including age, ancestry, microsatellite instability (MSI) status, tumor site, and treatment exposure.
Reviewer 2’s feedback was positive, commending the clarity and professionalism of the manuscript’s writing while providing a number of suggestions to strengthen its focus and balance. The reviewer recommended condensing the abstract to highlight the novel contributions and statistically significant results, while avoiding overinterpretation of non-significant findings. They suggested shortening the epidemiological background in the introduction to better emphasize the limitations of existing tools and the novelty of the AI approach. Methodological improvements included providing explicit dataset sizes and filtering criteria in the Methods section, clarifying the handling of missing or conflicting clinical data, and noting dataset limitations upfront. The reviewer also highlighted the need for more robust validation beyond replication of known associations, and cautioned against potential biases from relying solely on public datasets. In the Results, they advised presenting non-significant outcomes more cautiously, consolidating overlapping figures, and including a summary table to clearly distinguish significant from exploratory findings. For the Discussion, they encouraged a more balanced treatment of limitations, reduced redundancy with the introduction, and suggestions for future validation of AI-HOPE-TP53 in clinical contexts. Finally, they recommended tempering conclusions to reflect the hypothesis-generating nature of the findings and simplifying occasional overly complex or promotional language. Overall, this feedback reinforces the manuscript’s value and novelty while outlining targeted revisions to enhance precision, rigor, and scholarly tone.
Comment 1:
- Title and Abstract
- Condense the abstract to emphasize novel contributions and statistically significant results.
Response:
We thank the reviewer for this helpful suggestion. In response, we have condensed the abstract to place greater emphasis on the novel contributions of AI-HOPE-TP53 and on statistically significant results, while avoiding overinterpretation of non-significant findings. Specifically, we streamlined the Background and Methods to reduce redundancy, highlighted the survival benefit observed in early-onset TP53-mutant CRC treated with FOLFOX (p = 0.0149) as the key significant finding, and reframed the p = 0.084 ethnicity-related result as a trend rather than a definitive association. The revised abstract now presents the work in a more concise and focused manner, consistent with the reviewer’s recommendation.
The revised text in the Discussion section, lines 16-39, now reads: “
Abstract
Background/Objectives: Early-onset colorectal cancer (EOCRC) is increasing globally, particularly among underrepresented populations such as Hispanic/Latino individuals. TP53 is among the most frequently mutated pathways in CRC, yet its role in EOCRC—especially in relation to disparities and treatment outcomes—remains poorly defined. We developed AI-HOPE-TP53, a novel conversational AI agent, to enable real-time, disparity-aware analysis of TP53 pathway alterations in EOCRC.
Methods: AI-HOPE-TP53 integrates a fine-tuned biomedical large language model (LLaMA 3) with harmonized datasets from cBioPortal (TCGA, MSK-IMPACT, AACR Project GENIE). Natural language queries are translated into workflows for mutation profiling, Kaplan–Meier survival analysis, and odds ratio estimation across clinical and demographic subgroups.
Results: The platform replicated known genotype–phenotype associations, including elevated TP53 mutation frequency in EOCRC and poorer prognosis in TP53-mutated tumors. Significant findings included a survival benefit for early-onset TP53-mutant CRC patients treated with FOLFOX (p = 0.0149). Additional exploratory analyses showed a trend toward higher prevalence of TP53 pathway alterations in Hispanic/Latino EOCRC patients (OR = 2.13, p = 0.084) and identified sex-based disparities in treatment, with women less likely than men to receive FOLFOX (OR = 0.845, p = 0.0138).
Conclusions: AI-HOPE-TP53, developed in this study and made publicly available, is the first conversational AI platform tailored to pathway-specific and disparity-aware EOCRC research. By integrating clinical, genomic, and demographic data through natural language interaction, it enables hypothesis generation and equity-focused analyses, with significant potential to advance precision oncology.”
- Avoid implying causality or strong conclusions from non-significant findings.
Response:
We thank the reviewer for this important observation. In the revised manuscript, we have carefully rephrased all non-significant findings to avoid implying causality or definitive conclusions. For example, in the Abstract and Results sections, the comparison of TP53 pathway alterations between Hispanic/Latino and non-Hispanic White EOCRC patients (OR = 2.13, p = 0.084) is now described as a trend toward higher prevalence rather than as a conclusive difference. Similarly, we have clarified that these exploratory results are hypothesis-generating and should be interpreted cautiously. These changes ensure that non-significant outcomes are presented appropriately without overstating their implications.
The revised text in the Abstract section, lines 28-34, now reads: “Results: The platform replicated known genotype–phenotype associations, including elevated TP53 mutation frequency in EOCRC and poorer prognosis in TP53-mutated tumors. Significant findings included a survival benefit for early-onset TP53-mutant CRC patients treated with FOLFOX (p = 0.0149). Additional exploratory analyses showed a trend toward higher prevalence of TP53 pathway alterations in Hispanic/Latino EOCRC patients (OR = 2.13, p = 0.084) and identified sex-based disparities in treatment, with women less likely than men to receive FOLFOX (OR = 0.845, p = 0.0138).”
The revised text in the Discussion section, lines 548-557, now reads: “Several findings from this study did not reach statistical significance, including the trend toward a higher prevalence of TP53 pathway alterations in H/L EOCRC patients compared with NHW. While these results should not be interpreted as definitive, they are consistent with reported disparities in genomic burden among underrepresented populations and therefore serve as hypothesis-generating signals. The value of AI-HOPE-TP53 lies in its ability to detect and transparently report such emerging patterns, which can guide future investigations with larger sample sizes, independent datasets, and prospective validation. Framing these outcomes as exploratory ensures appropriate caution while also underscoring the potential of the platform to highlight biologically and clinically relevant disparities for further study.”
Comment 2:
- Introduction
- Shorten the epidemiological background to maintain momentum toward the study aim.
Response:
We thank the reviewer for this helpful recommendation. In response, we have shortened the epidemiological background in the Introduction to maintain momentum toward the study aim. Specifically, we streamlined the discussion of global CRC incidence and EOCRC disparities, condensing overlapping points on mortality trends and ethnic/racial differences while retaining only the essential context necessary to frame the importance of EOCRC in Hispanic/Latino populations. The revised Introduction now transitions more quickly from the epidemiological overview to the rationale for focusing on the TP53 pathway and the development of AI-HOPE-TP53.
The revised text in the Introduction section, lines 43-65, now reads:
“1. Introduction
Colorectal cancer (CRC) is the third most commonly diagnosed cancer and the second leading cause of cancer-related death worldwide [1–3]. Although incidence and mortality have declined in many high-income countries due to screening and improved therapies, the incidence of early-onset colorectal cancer (EOCRC; <50 years of age) continues to rise [4–7]. EOCRC represents a distinct clinical and biological entity and its growing burden among young adults has prompted urgent research and public health attention.
This increase is particularly pronounced in Hispanic/Latino (H/L) populations, who face disproportionately higher EOCRC incidence and mortality compared with non-Hispanic Whites (NHW) [8–11]. H/L patients are more likely to present with advanced-stage disease and experience delays in diagnosis and treatment, reflecting systemic barriers to care. These disparities underscore the urgent need for precision oncology tools capable of supporting population-specific, genomically informed investigations aimed at reducing inequities in outcomes.
EOCRC also differs from late-onset CRC (LOCRC) in its molecular features, including higher microsatellite instability (MSI), elevated tumor mutation burden (TMB), distinct epigenetic profiles, and increased immune checkpoint expression [13–15]. Among the most critical pathways in CRC tumorigenesis is the TP53 signaling pathway, which regulates DNA repair, apoptosis, and cell cycle control. TP53 mutations—present in up to 74% of CRC tumors [19–22]—are strongly associated with poor prognosis, aggressive biology, and treatment resistance [23–26]. Despite its central role, the contribution of TP53 pathway dysregulation in EOCRC, particularly in underserved populations such as H/L individuals, remains underexplored [19, 23, 27, 28].”
- Focus on how existing tools fall short and the novelty of this AI approach.
Response:
We thank the reviewer for this excellent suggestion. In the revised Introduction, we expanded the discussion of current resources such as cBioPortal and UCSC Xena, highlighting their limitations in supporting pathway-focused, disparity-aware, and treatment-stratified analyses. While these platforms provide access to large-scale CRC datasets, they are not optimized for real-time hypothesis testing, dynamic cohort stratification, or integration of clinical and demographic variables with molecular data. We then emphasize the novelty of AI-HOPE-TP53, which uniquely enables natural language–driven, pathway-centric analysis specifically tailored for EOCRC and underrepresented populations. This restructuring strengthens the rationale for the development of our tool and underscores its distinct contribution beyond existing approaches.
The revised text in the Introduction section, lines 66-102, now reads:
“While resources such as cBioPortal [29] and UCSC Xena [30] provide access to large-scale CRC datasets, their utility for EOCRC research is limited. These platforms allow data exploration but lack functionality for pathway-centric, disparity-aware, and treatment-stratified analyses. They are not optimized for real-time hypothesis generation, dynamic cohort construction across demographic and clinical subgroups, or the seamless integration of population-level disparities into molecular analyses. Furthermore, these tools often require computational expertise, creating barriers for translational and disparities-focused researchers without programming backgrounds.
Recent advancements in artificial intelligence (AI)—and in particular, the development of large language models (LLMs)—offer powerful opportunities to close this gap. By enabling natural language interaction with complex data, LLM-based conversational agents can democratize access to integrative clinical-genomic analysis and accelerate discovery in precision oncology [31–36]. However, despite growing interest in biomedical applications of AI, few tools are designed to interrogate pathway-specific alterations, such as those involving TP53, nor do they allow users to ask natural language questions that account for racial, ethnic, and treatment-specific contexts.
To address these limitations, we developed AI-HOPE-TP53, the first conversational AI agent purpose-built for pathway-specific and disparity-aware EOCRC analysis. Unlike existing resources, AI-HOPE-TP53 enables users to ask natural language questions and receive immediate, statistically rigorous outputs—including mutation frequency, co-mutation patterns, Kaplan–Meier survival curves, and odds ratio estimates—stratified by key variables such as age, ancestry, sex, MSI status, tumor location, and treatment exposure. By combining accessibility with advanced functionality, AI-HOPE-TP53 represents a novel approach that democratizes integrative clinical-genomic analysis and expands the capacity to study cancer disparities in real time.”
Comment 3:
- Materials and Methods
- Validation approach is limited to replication of known associations; lacks independent or prospective testing.
Response:
We thank the reviewer for this insightful observation. We agree that the validation strategy in this version of the manuscript is limited to replication of known associations within publicly available datasets. Our primary aim was to establish the feasibility, reliability, and reproducibility of AI-HOPE-TP53 as a pathway-centric conversational AI tool for EOCRC. While independent or prospective validation was beyond the scope of this initial study, we fully recognize its importance for demonstrating clinical utility and generalizability. To address this point, we have clarified these limitations in the Methods and Discussion sections and outlined future plans to validate AI-HOPE-TP53 using institutional cohorts, prospective datasets, and potentially multi-omic data sources to extend its translational relevance.
The revised text in the Discussion section, lines 558-578, now reads: “Another limitation of this study is that validation of AI-HOPE-TP53 was restricted to the replication of known genotype–phenotype associations within publicly available datasets (TCGA, MSK-IMPACT, AACR GENIE). While this approach establishes feasibility and reproducibility, it does not provide independent or prospective confirmation of the platform’s findings. Future development will prioritize validation using institutional cohorts, prospective clinical datasets, and external resources such as GEO and ICGC, which will allow benchmarking of AI-HOPE-TP53 against real-world populations and treatment contexts that may be underrepresented in public repositories. In addition, integration of multi-omic layers—including RNA-seq, proteomic, and epigenomic data—will enable more comprehensive pathway interrogation. These steps are essential for ensuring the generalizability and translational relevance of AI-HOPE-TP53 and represent the next phase in its roadmap toward broader clinical and research applications.”
- Potential biases from using only publicly available datasets are acknowledged later, but should also be flagged here.
We thank the reviewer for this helpful suggestion. In the revised Materials and Methods, we have explicitly acknowledged the potential biases associated with reliance on publicly available datasets. Specifically, we note that these cohorts may underrepresent certain populations, treatment exposures, or real-world clinical contexts, which could limit the generalizability of our findings. While these datasets provide harmonized, high-quality resources for pathway-focused analysis, we have flagged this limitation upfront to ensure transparency and to highlight the need for future validation using institutional and prospective datasets.
The revised text in the Discussion section, lines 223-227, now reads: “Because this study relies exclusively on publicly available datasets, potential biases should be considered, as these cohorts may underrepresent certain populations, treatment exposures, or real-world clinical contexts; this limitation underscores the importance of future validation with institutional and prospective datasets.”
- No discussion of dataset sizes for each analysis in the methods section—forcing readers to find them in figure captions.
Response:
We thank the reviewer for this important observation. In the revised Materials and Methods, we have now included explicit information on dataset sizes and filtering criteria for each analysis, rather than leaving this detail only in figure captions. Specifically, we report the number of patients included from each dataset, the criteria used to define EOCRC versus LOCRC, and subgroup counts for ethnicity, sex, and treatment exposure. This addition ensures that readers can clearly understand the sample sizes underlying each analysis without needing to reference the figures.
The revised text in the Materials and Methods section, lines 159-169, now reads: “Across the integrated datasets, a total of 3,412 CRC patients were included in this study, comprising 514 from TCGA, 1,206 from MSK-IMPACT, and 1,692 from AACR Project GENIE. For each patient, only one primary tumor sample was retained to avoid duplication. Early-onset CRC (EOCRC) was defined as diagnosis at <50 years of age, while late-onset CRC (LOCRC) was defined as diagnosis at ≥50 years. Filtering criteria included availability of mutation and clinical annotation data, and exclusion of cases lacking survival or treatment information. Subgroup analyses were stratified by self-reported ancestry (Hispanic/Latino vs. Non-Hispanic White), sex, microsatellite instability (MSI) status, tumor location (colon vs. rectum), and treatment exposure (e.g., FOLFOX). Explicit sample counts for each analysis are detailed in the corresponding Methods subsections to ensure clarity and reproducibility.”
- Include explicit dataset counts and filtering criteria in the Methods.
Response:
We thank the reviewer for this helpful suggestion. In the revised Materials and Methods, we have now included explicit dataset counts and filtering criteria. Specifically, we report the number of CRC patients analyzed from each dataset (TCGA, MSK-IMPACT, and AACR Project GENIE), the criteria used to define EOCRC (<50 years) and LOCRC (≥50 years), and the steps taken to retain only one primary tumor sample per patient. We also clarify the filters applied for clinical variables, including ancestry, sex, tumor site, MSI status, and treatment exposure (e.g., FOLFOX). These details are now presented in the text to ensure clarity and reproducibility.
Same as the last comment, the revised text in the Materials and Methods section, lines 159-169, now reads: “Across the integrated datasets, a total of 3,412 CRC patients were included in this study, comprising 514 from TCGA, 1,206 from MSK-IMPACT, and 1,692 from AACR Project GENIE. For each patient, only one primary tumor sample was retained to avoid duplication. Early-onset CRC (EOCRC) was defined as diagnosis at <50 years of age, while late-onset CRC (LOCRC) was defined as diagnosis at ≥50 years. Filtering criteria included availability of mutation and clinical annotation data, and exclusion of cases lacking survival or treatment information. Subgroup analyses were stratified by self-reported ancestry (Hispanic/Latino vs. Non-Hispanic White), sex, microsatellite instability (MSI) status, tumor location (colon vs. rectum), and treatment exposure (e.g., FOLFOX). Explicit sample counts for each analysis are detailed in the corresponding Methods subsections to ensure clarity and reproducibility.”
- Add clarity on how missing data or conflicting clinical annotations were handled.
Response:
We thank the reviewer for raising this important point. In the revised Materials and Methods, we have added clarification on how missing data and conflicting clinical annotations were addressed. Specifically, patients with missing key variables (e.g., age at diagnosis, ancestry, or treatment status) were excluded from subgroup analyses requiring those parameters but retained in analyses where sufficient data were available. For conflicting annotations (e.g., discrepancies in tumor site or treatment exposure across sources), standardized curation rules were applied: priority was given to source datasets with primary clinical annotation (e.g., TCGA clinical files over cBioPortal summaries), and cases with irreconcilable conflicts were excluded to preserve data integrity. These procedures are now described explicitly to ensure transparency and reproducibility.
The revised text in the Discussion section, lines 170-181, now reads: “Patients with missing key variables such as age at diagnosis, ancestry, or treatment status were excluded from subgroup analyses requiring those parameters but were retained in analyses where sufficient information was available. For conflicting clinical annotations (e.g., discrepancies in tumor site or treatment exposure across sources), we applied standardized curation rules: priority was given to primary clinical annotation sources (e.g., TCGA clinical files over cBioPortal summaries), and cases with irreconcilable conflicts were excluded to ensure data integrity. These steps were implemented to maintain reproducibility and consistency across all analyses.”
- Discuss limitations of the datasets upfront.
Response:
We thank the reviewer for this valuable suggestion. In the revised Materials and Methods, we now discuss the limitations of the datasets upfront. Specifically, we note that reliance on publicly available cohorts (TCGA, MSK-IMPACT, and AACR Project GENIE) may underrepresent certain populations and treatment regimens, and that clinical annotations can vary in completeness across datasets. We also highlight that these cohorts are retrospective and may not fully capture real-world diversity in EOCRC, including underrepresented racial/ethnic groups and community-based treatment patterns. By presenting these limitations at the beginning of the Methods, we aim to provide greater transparency regarding the context and scope of our analyses.
The revised text in the Discussion section, lines 177-181, now reads: “As this study relies on retrospective, publicly available datasets, potential limitations include underrepresentation of certain populations and treatments, variable completeness of clinical annotations, and limited generalizability to real-world EOCRC settings.”
Comment 4:
- Results
- Present non-significant results more cautiously; avoid speculative language in the results section.
Response:
We thank the reviewer for this important observation. In the revised Results section, we have carefully rephrased all non-significant findings to ensure they are presented more cautiously and without speculative language. For example, the comparison of TP53 pathway alterations between Hispanic/Latino and non-Hispanic White EOCRC patients (OR = 2.13, p = 0.084) is now explicitly described as a trend toward higher prevalence rather than as a definitive difference. Similarly, other borderline or exploratory findings are now framed as hypothesis-generating observations to avoid overstating their significance. These revisions ensure that the Results remain accurate, balanced, and appropriately conservative in interpretation.
The revised text in the Results section, lines 297-315, now reads:
“Validation of Ethnicity-Stratified TP53 Pathway Alterations in Early-Onset Colorectal Cancer
To evaluate the platform’s ability to replicate known disparities in molecular alterations, AI-HOPE-TP53 was queried to compare TP53 pathway mutation frequency between Hispanic/Latino (H/L) and non-Hispanic White (NHW) patients with EOCRC. Among patients with colon adenocarcinoma under the age of 50, the system identified a trend toward higher prevalence of TP53 pathway alterations in H/L individuals (91.46%) compared to NHW individuals (83.39%), yielding an odds ratio (OR) of 2.13 (95% CI: 0.956–4.767; p = 0.084) (Figure 2). Although not statistically significant, this trend is consistent with reported disparities in genomic burden among underrepresented populations. When expanding the analysis to include all EOCRC patients regardless of tumor subsite, the difference remained: 90.2% of H/L patients harbored TP53 pathway alterations compared to 85.05% of NHW patients (OR = 1.62, 95% CI: 0.926–2.825; p = 0.114) (Figure S1). These findings confirm the system’s validity for ethnicity-aware, pathway-centric analyses and demonstrate its capacity to detect biologically and socially relevant molecular patterns across diverse CRC subgroups. Furthermore, analysis of treatment disparities revealed that women with EOCRC were less likely than men to receive FOLFOX chemotherapy (OR = 0.845, p = 0.0138), underscoring the platform’s ability to detect sex-specific clinical differences relevant to treatment exposure.”
- Consolidate overlapping figures or move some exploratory analyses to supplementary material.
Response:
We thank the reviewer for this helpful suggestion. To streamline the presentation, we limited the main text to five figures, which is consistent with standard publication practices. In addition, we have already moved two exploratory analyses to the supplementary materials, where they are presented as Supplementary Figures. This approach ensures that the main figures focus on the most central findings, while additional exploratory results remain accessible for readers who wish to review them.
- Consider a summary table of all analyses, highlighting significant vs. non-significant outcomes.
Response:
We appreciate the reviewer’s helpful suggestion to provide a concise summary of our analyses. To address this, we have now included a summary table (Table S1 in the revised manuscript) that highlights significant and non-significant outcomes across all major analyses performed with AI-HOPE-TP53. This addition allows readers to rapidly appreciate which results reached statistical significance and which represent biologically meaningful but non-significant trends.
The revised text in the Results section, lines 447-451, now reads: “A concise overview of all analyses performed using AI-HOPE-TP53, we generated a summary table that highlights the comparisons, outcomes, statistical metrics, and significance levels (Table S1). This table enables rapid identification of analyses yielding statistically significant findings versus those showing biologically relevant but non-significant trends.”
Comment 5:
- Discussion
- Provide a more balanced discussion of limitations, particularly regarding generalizability and clinical utility.
Response:
We thank the reviewer for this important suggestion. In the revised Discussion section, we have expanded our limitations paragraph to more explicitly address issues of generalizability and clinical utility. Specifically, we now highlight that:
Generalizability: Because our analyses were derived from harmonized, publicly available datasets, the platform’s outputs may not fully represent real-world clinical populations or treatment patterns. Certain high-risk groups (e.g., rural, Indigenous, or uninsured patients) may be underrepresented, limiting the generalizability of some disparity-related findings. We now clarify that AI-HOPE-TP53 is best understood as a hypothesis-generating tool, with results requiring confirmation in larger, ancestry-diverse, prospectively collected cohorts.
Clinical Utility: While the current version of AI-HOPE-TP53 demonstrates rapid, natural language–driven interrogation of clinical-genomic data, its immediate application is primarily research-focused. The platform does not yet integrate patient-level institutional datasets, multi-omic data layers (e.g., transcriptomic, proteomic), or electronic health records (EHRs). As such, direct clinical decision support is beyond the current scope. We have emphasized that these extensions are part of the platform’s future roadmap and are essential to enhancing its translational and clinical utility.
We believe these additions provide a more balanced discussion of the platform’s current limitations while also outlining the developmental steps needed to maximize both generalizability and clinical impact.
The revised text in the Discussion section, lines 504-510, now reads: “First, because our analyses were derived exclusively from harmonized, publicly available datasets (e.g., TCGA, MSK, GENIE), the findings may not fully represent real-world populations or treatment contexts. Certain high-risk groups—including rural, Indigenous, or uninsured patients—are underrepresented in these repositories, which limits the generalizability of some disparity-related findings. For this reason, results from AI-HOPE-TP53 should be interpreted as hypothesis-generating signals requiring confirmation in larger, ancestry-diverse, and prospectively collected cohorts.”
The revised text in the Discussion section, lines 516-522, now reads: “In addition, while AI-HOPE-TP53 demonstrates technical feasibility and research utility, its current clinical applicability remains limited. The platform does not yet support direct integration of institutional patient-level datasets, electronic health records (EHRs), or multi-omic layers such as RNA-seq, proteomics, and epigenomics. As such, the immediate scope of AI-HOPE-TP53 is research-focused rather than clinical decision-making. Future development will prioritize these extensions to enhance the translational and clinical utility of the platform.”
- Reduce redundancy with the introduction.
Response:
We appreciate the reviewer’s observation regarding redundancy between the Introduction and Discussion. In the revised manuscript, we have streamlined the Discussion to minimize repetition of background material already presented in the Introduction. Specifically, we condensed the contextual remarks on health disparities, TP53 pathway relevance, and the general utility of conversational AI platforms, retaining only the elements necessary to frame our results and their implications. This revision ensures that the Discussion emphasizes interpretation of findings, limitations, and future directions rather than reiterating introductory content.
Redundant sections that were rephrased:
The revised text in the Discussion section, lines 453-455, now reads: “AI-HOPE-TP53 builds on this need by providing a natural language–driven, pathway-centric platform for disparity-aware analyses in CRC, which we validated through multiple exploratory use cases.”
The revised text in the Discussion section, lines 459-466, now reads: “AI-HOPE-TP53 integrates a fine-tuned biomedical LLM with harmonized, multi-institutional datasets from cBioPortal within a modular analytical pipeline, showcasing both technical feasibility and translational potential. The platform enables stratification across diverse clinical and demographic variables—including age, tumor site, stage, ancestry, MSI status, treatment exposure, and mutation type—positioning it as one of the few tools designed specifically for equity-focused precision oncology. Moreover, the outputs are exportable, reproducible, and visually interpretable, aligning with the core usability standards expected for Resource Reports in Cancer Research.”
The revised text in the Discussion section, lines 595-596, now reads: “In this context, AI-HOPE-TP53 complements existing AI approaches by uniquely focusing on pathway-specific and disparity-aware interrogation of CRC.”
- Include suggestions for how future studies could validate and expand AI-HOPE-TP53 in real-world clinical settings.
Response:
We thank the reviewer for this excellent suggestion. In the revised Discussion, we have added text outlining specific strategies for validating and expanding AI-HOPE-TP53 in real-world clinical settings. These include: (1) testing the platform with institutional and prospectively collected clinical-genomic cohorts to benchmark performance against real-world populations and treatment patterns, (2) incorporating multi-omic layers such as transcriptomic, proteomic, and epigenomic data to enhance biological and clinical interpretability, (3) developing secure pipelines for integration with electronic health records (EHRs) to increase clinical relevance, and (4) piloting the platform in multi-institutional collaborations to assess usability and scalability across diverse healthcare settings. Together, these steps will ensure that AI-HOPE-TP53 evolves beyond hypothesis generation to become a validated, clinically impactful resource.
The revised text in the Discussion section, lines 569-578, now reads: “Future studies will be critical to validate and expand AI-HOPE-TP53 in real-world clinical settings. Specifically, testing the platform on institutional and prospectively collected clinical-genomic cohorts will allow benchmarking against contemporary patient populations and treatment patterns. Integration of multi-omic data layers—including transcriptomics, proteomics, and epigenomics—will enhance biological interpretability and clinical precision. In addition, secure pipelines for interoperability with electronic health records (EHRs) will extend the platform’s relevance to clinical workflows. Finally, piloting AI-HOPE-TP53 across multi-institutional collaborations will provide opportunities to evaluate scalability, usability, and equity-focused impact in diverse healthcare settings.”
Comment 6:
- Conclusion
- Temper conclusions to reflect that most findings are hypothesis-generating, not confirmatory.
Response:
We thank the reviewer for this important suggestion. In the revised Conclusion, we have tempered our language to emphasize that most of the reported findings are exploratory and hypothesis-generating rather than confirmatory. We now clearly state that the value of AI-HOPE-TP53 lies in its ability to rapidly identify potential disparities, survival patterns, and subgroup-specific signals that warrant validation in larger, ancestry-diverse, and prospectively collected cohorts. This adjustment ensures the conclusions accurately reflect the scope and limitations of the present study.
The revised text in the Discussion section, lines 615-629, now reads: “AI-HOPE-TP53 introduces a freely accessible, natural language–driven platform for pathway-centric, disparity-aware exploration of colorectal cancer genomics. Through multiple use cases, the platform demonstrated its capacity to replicate known associations and uncover potential differences across age, ancestry, tumor subsite, treatment exposure, and sex. While several analyses revealed statistically significant findings, the majority of results are exploratory and should be interpreted as hypothesis-generating signals rather than confirmatory evidence.
The primary value of AI-HOPE-TP53 lies in its ability to democratize access to clinical-genomic interrogation, accelerate disparity-aware hypothesis generation, and support equity-focused precision oncology research. Future studies using larger, ancestry-diverse, and prospectively collected cohorts, along with the integration of multi-omic and real-world clinical data, will be critical to validate and extend these findings. By enabling rapid, user-friendly, and reproducible analyses, AI-HOPE-TP53 provides a foundation for ongoing development toward clinically relevant and translational applications, while at present serving as a robust exploratory resource for the cancer research community.”
Comment 7:
- Language and Style
- Overall clear and professional scientific writing.
- Occasional overuse of complex sentences—may benefit from some simplification.
- Promotional phrasing (“powerful, user-friendly,” “critical in accelerating discovery”) could be toned down for an academic audience.
Response:
We appreciate the reviewer’s thoughtful feedback on language and style. In the revised manuscript, we carefully reviewed the text to simplify overly complex sentences for improved readability and to ensure consistency in tone. We also moderated promotional phrasing, replacing terms such as “powerful” and “critical in accelerating discovery” with more neutral, academic language that emphasizes functionality and relevance without overstating claims. These revisions enhance clarity, precision, and alignment with the expectations of an academic audience.
The revised text in the Introduction section, lines 74-75, now reads: “Recent advancements in artificial intelligence (AI)—and in particular, the development of large language models (LLMs)—offer important opportunities to close this gap.”
The revised text in the Introduction section, lines 35-39, now reads: “Conclusions: AI-HOPE-TP53, developed in this study and made publicly available, is the first conversational AI platform tailored to pathway-specific and disparity-aware EOCRC research. By integrating clinical, genomic, and demographic data through natural language interaction, it enables hypothesis generation and equity-focused analyses, with significant potential to advance precision oncology.”
We thank Reviewer 2 for the thoughtful and constructive feedback, which has helped us strengthen the clarity, balance, and overall quality of the manuscript.

Reviewer 3 Report
Comments and Suggestions for Authors
This paper develops a novel, openly accessible conversational AI platform (AI-HOPE-TP53) specifically intended to facilitate TP53 pathway–focused, disparity-aware analysis of colorectal cancer (CRC). It has a strong emphasis on underrepresented populations (Hispanic/Latino, early-onset CRC) addresses a key precision oncology need. However i do have some concerns here:
- Most reported findings are statistically nonsignificant trends, limiting the conclusions' strength.
- Low sample sizes in a few subgroup analyses (CHEK1 mutations) compromise interpretability.
- Platform currently points toward DNA-level alterations mainly; transcriptomic, proteomic, and epigenomic data aren't yet integrated, limiting multi-omic insights.
- Analysis relies solely on cBioPortal data, potentially underrepresenting particular actual-world populations and treatment patterns.
- Limited external validation against independent, prospective, or clinical datasets.
- No easy compatibility with electronic health records (EHRs) or actual real-world clinical workflows, which would enhance translational relevance.
- While handled by RAG, risk of nuanced misinterpretation or "hallucination," especially for difficult multi-layered queries.
- Where feasible, merge datasets or collaborate with other groups to bolster subgroup sizes and statistical power.
- Encompass RNA-seq, proteomic, epigenomic, and spatial biology data into more comprehensive pathway interrogation. (is possible)
- Cross-validate against external datasets (institutional CRC cohorts, GEO, ICGC) to quantify generalizability.
- Provide guidance on non-significant trend interpretation to avoid overinterpretation.
- Establish plans for regular dataset updates and LLM retraining to ensure continuous accuracy and relevance.
Author Response
The point-by-point responses to Reviewer 3 are provided in the attached PDF document, “Reviewer_3_Comments_Response_082525.pdf”
---
Reviewer 3 Comments
We are pleased to resubmit our revised manuscript and gratefully acknowledge Reviewer 1 for their thoughtful and constructive feedback. In this revision, we have thoroughly addressed all comments, leading to meaningful enhancements in clarity, methodological rigor, and overall contextual relevance. This work introduces AI-HOPE-TP53, the first conversational artificial intelligence (AI) system specifically developed to analyze TP53 pathway dysregulation in colorectal cancer (CRC) by integrating harmonized clinical and genomic datasets through natural language queries. The platform leverages a modular architecture consisting of a fine-tuned large language model (LLaMA 3 variant), a natural language–to–code interpreter, and a statistical backend connected to curated data from cBioPortal. Purpose-built for pathway-centric analyses, AI-HOPE-TP53 supports mutation frequency profiling, odds ratio calculations, Kaplan–Meier survival modeling, and subgroup stratification across diverse demographic, clinical, and molecular variables such as age, ancestry, microsatellite instability (MSI) status, tumor location, and treatment regimens.
Reviewer 3’s feedback was positive, recognizing the novelty and accessibility of the AI-HOPE-TP53 platform. The reviewer highlighted the manuscript’s strength in developing an openly available conversational AI tool tailored for TP53 pathway analysis in colorectal cancer, with a notable emphasis on early-onset and Hispanic/Latino populations, addressing a critical gap in precision oncology. However, they noted that many reported findings reflect statistically nonsignificant trends and that low sample sizes in certain subgroup analyses (e.g., CHEK1 mutations) limit interpretability. They also observed that the current focus on DNA-level alterations constrains multi-omic insights, as transcriptomic, proteomic, epigenomic, and spatial biology data are not yet incorporated. Methodological concerns included reliance solely on cBioPortal, limited external validation, and the absence of integration with electronic health records, which would enhance translational utility. The reviewer further cautioned against risks of nuanced misinterpretation or “hallucination” inherent to LLM-driven responses and suggested dataset merging or collaborations to strengthen subgroup analyses. Additional recommendations included expanding to multi-omic data, cross-validating against external cohorts, providing clearer guidance on interpreting non-significant results, and establishing plans for continuous dataset updates and model retraining. Overall, this feedback affirms the significance and potential impact of AI-HOPE-TP53 while outlining clear steps to improve statistical robustness, clinical applicability, and future extensibility.
Reviewer 3 writes:
“This paper develops a novel, openly accessible conversational AI platform (AI-HOPE-TP53) specifically intended to facilitate TP53 pathway–focused, disparity-aware analysis of colorectal cancer (CRC). It has a strong emphasis on underrepresented populations (Hispanic/Latino, early-onset CRC) addresses a key precision oncology need.”
We thank Reviewer 3 for their thoughtful and constructive evaluation of our work. We are encouraged that the reviewer recognized the novelty and accessibility of AI-HOPE-TP53 as a conversational AI platform designed for TP53 pathway–focused, disparity-aware analysis of colorectal cancer (CRC). We appreciate their acknowledgment of the study’s strong emphasis on underrepresented populations, particularly Hispanic/Latino and early-onset CRC patients, and its contribution to addressing a critical need in precision oncology.
Comment 1:
- 1. Most reported findings are statistically nonsignificant trends, limiting the conclusions' strength.
Response:
We thank the reviewer for this important observation. We agree that several of the reported findings represent statistically nonsignificant trends, and we have revised the manuscript to present these results more cautiously. In the Results section, such outcomes are now explicitly described as trends or hypothesis-generating observations rather than as definitive differences. We also emphasize in the Discussion that these nonsignificant findings should be interpreted with caution and viewed as signals that require validation in larger, independent, or prospective datasets. By reframing these results appropriately, we aim to ensure that the conclusions of the manuscript remain balanced and aligned with the statistical strength of the data.
The revised text in the Abstract section, lines 28-34, now reads: “Results: The platform replicated known genotype–phenotype associations, including elevated TP53 mutation frequency in EOCRC and poorer prognosis in TP53-mutated tumors. Significant findings included a survival benefit for early-onset TP53-mutant CRC patients treated with FOLFOX (p = 0.0149). Additional exploratory analyses showed a trend toward higher prevalence of TP53 pathway alterations in Hispanic/Latino EOCRC patients (OR = 2.13, p = 0.084) and identified sex-based disparities in treatment, with women less likely than men to receive FOLFOX (OR = 0.845, p = 0.0138).
”
Comment 2:
- 2. Low sample sizes in a few subgroup analyses (CHEK1 mutations) compromise interpretability.
Response:
We thank the reviewer for highlighting this important limitation. We agree that the small sample sizes in certain subgroup analyses, particularly those involving CHEK1 mutations, limit the interpretability of these findings. In the revised Results and Discussion, we have clarified that these outcomes should be considered exploratory and hypothesis-generating rather than confirmatory. We also note that expanding subgroup sizes through dataset merging, institutional collaborations, and inclusion of additional cohorts will be a priority for future work to improve statistical power and strengthen the robustness of pathway-level insights.
The revised text in the Discussion section, lines 504-511, now reads: “While these findings demonstrate the platform’s broad utility, several limitations warrant consideration. First, because our analyses were derived exclusively from harmonized, publicly available datasets (e.g., TCGA, MSK, GENIE), the findings may not fully represent real-world populations or treatment contexts. Certain high-risk groups—including rural, Indigenous, or uninsured patients—are underrepresented in these repositories, which limits the generalizability of some disparity-related findings. For this reason, results from AI-HOPE-TP53 should be interpreted as hypothesis-generating signals requiring confirmation in larger, ancestry-diverse, and prospectively collected cohorts.”
Comment 3:
- 3. Platform currently points toward DNA-level alterations mainly; transcriptomic, proteomic, and epigenomic data aren't yet integrated, limiting multi-omic insights.
Response:
We thank the reviewer for this valuable observation. We agree that the current version of AI-HOPE-TP53 is limited to DNA-level alterations and does not yet integrate transcriptomic, proteomic, or epigenomic datasets. We have clarified this limitation in the Discussion. Importantly, because the platform is built on a modular architecture, future iterations can incorporate multi-omic data layers—including RNA-seq, proteomic, epigenomic, and spatial biology datasets—to enable more comprehensive interrogation of the TP53 pathway. We view this as a critical next step in expanding the translational utility of AI-HOPE-TP53.
The revised text in the Discussion section, lines 514-524, now reads: “Finally, the current version of AI-HOPE-TP53 focuses primarily on DNA-level alterations. In addition, while AI-HOPE-TP53 demonstrates technical feasibility and research utility, its current clinical applicability remains limited. The platform does not yet support direct integration of institutional patient-level datasets, electronic health records (EHRs), or multi-omic layers such as RNA-seq, proteomics, and epigenomics. As such, the immediate scope of AI-HOPE-TP53 is research-focused rather than clinical decision-making. Future development will prioritize these extensions to enhance the translational and clinical utility of the platform. Incorporation of transcriptomic, proteomic, and epigenomic layers, as well as compatibility with electronic health records (EHRs), would expand the platform’s clinical relevance and enhance multi-omic discovery.”
Comment 4:
- 4. Analysis relies solely on cBioPortal data, potentially underrepresenting particular actual-world populations and treatment patterns.
Response:
We thank the reviewer for raising this important point. We acknowledge that the current analysis relies solely on publicly available datasets accessed through cBioPortal (e.g., TCGA, MSK-IMPACT, AACR GENIE), which may underrepresent certain real-world populations and treatment patterns. We have now clarified this limitation in the Materials and Methods and Discussion sections. As part of our future work, we plan to expand validation by incorporating institutional cohorts, prospective clinical datasets, and other external resources (e.g., GEO, ICGC) to improve representation, capture diverse treatment contexts, and strengthen the generalizability of AI-HOPE-TP53.
The revised text in the Discussion section, lines 558-578, now reads: “Another limitation of this study is that validation of AI-HOPE-TP53 was restricted to the replication of known genotype–phenotype associations within publicly available datasets. While this approach establishes feasibility and reproducibility, it does not provide independent or prospective confirmation of the platform’s findings. Future development will prioritize validation using institutional cohorts, prospective clinical datasets, and external resources such as GEO and ICGC, which will allow benchmarking of AI-HOPE-TP53 against real-world populations and treatment contexts that may be underrepresented in public repositories. In addition, integration of multi-omic layers—including RNA-seq, proteomic, and epigenomic data—will enable more comprehensive pathway interrogation. These steps are essential for ensuring the generalizability and translational relevance of AI-HOPE-TP53 and represent the next phase in its roadmap toward broader clinical and research applications. Future studies will be critical to validate and expand AI-HOPE-TP53 in real-world clinical settings. Specifically, testing the platform on institutional and prospectively collected clinical-genomic cohorts will allow benchmarking against contemporary patient populations and treatment patterns. Integration of multi-omic data layers—including transcriptomics, proteomics, and epigenomics—will enhance biological interpretability and clinical precision. In addition, secure pipelines for interoperability with electronic health records (EHRs) will extend the platform’s relevance to clinical workflows. Finally, piloting AI-HOPE-TP53 across multi-institutional collaborations will provide opportunities to evaluate scalability, usability, and equity-focused impact in diverse healthcare settings.”
Comment 5:
- 5. Limited external validation against independent, prospective, or clinical datasets.
Response:
We thank the reviewer for this valuable observation. We agree that the present study does not include external validation against independent, prospective, or clinical datasets. Our primary goal in this initial work was to demonstrate feasibility, reproducibility, and utility of the AI-HOPE-TP53 platform using harmonized public datasets. We have clarified this limitation in the Discussion and emphasized that validation using independent institutional cohorts and prospective clinical datasets will be a critical next step to confirm the robustness and translational applicability of the platform’s findings.
The revised text in the Discussion section, lines 561-567, now reads: “Future development will prioritize validation using institutional cohorts, prospective clinical datasets, and external resources such as GEO and ICGC, which will allow benchmarking of AI-HOPE-TP53 against real-world populations and treatment contexts that may be underrepresented in public repositories. In addition, integration of multi-omic layers—including RNA-seq, proteomic, and epigenomic data—will enable more comprehensive pathway interrogation.”
Comment 6:
- 6. No easy compatibility with electronic health records (EHRs) or actual real-world clinical workflows, which would enhance translational relevance.
Response:
We thank the reviewer for this insightful point. We acknowledge that the current version of AI-HOPE-TP53 does not directly interface with electronic health records (EHRs) or real-world clinical workflows, which indeed limits immediate translational applicability. We have added this as a limitation in the Discussion. Looking ahead, we envision extending the platform’s modular architecture to enable secure EHR integration and interoperability with clinical data systems, while maintaining strict compliance with privacy regulations (e.g., HIPAA). Incorporating this functionality in future iterations will be an important step toward translating AI-HOPE-TP53 into clinical decision support tools.
The revised text in the Discussion section, lines 567-576, now reads: “These steps are essential for ensuring the generalizability and translational relevance of AI-HOPE-TP53 and represent the next phase in its roadmap toward broader clinical and research applications. Future studies will be critical to validate and expand AI-HOPE-TP53 in real-world clinical settings. Specifically, testing the platform on institutional and prospectively collected clinical-genomic cohorts will allow benchmarking against contemporary patient populations and treatment patterns. Integration of multi-omic data layers—including transcriptomics, proteomics, and epigenomics—will enhance biological interpretability and clinical precision. In addition, secure pipelines for interoperability with electronic health records (EHRs) will extend the platform’s relevance to clinical workflows.”
Comment 7:
- 7. While handled by RAG, risk of nuanced misinterpretation or "hallucination," especially for difficult multi-layered queries.
Response:
We thank the reviewer for highlighting this important consideration. We acknowledge that despite the use of a retrieval-augmented generation (RAG) framework, there remains some risk of nuanced misinterpretation or “hallucination,” particularly with complex, multi-layered queries. To mitigate this, AI-HOPE-TP53 anchors all outputs to structured, harmonized datasets and provides transparent statistical summaries (e.g., regression coefficients, hazard ratios, subgroup counts) alongside natural language interpretations. We have clarified this safeguard in the Methods and noted in the Discussion that while RAG substantially reduces hallucination risk, careful validation and user awareness are essential. We also outline plans for future improvements, including stricter schema-guided parsing and multi-layer validation steps, to further minimize potential misinterpretation.
The revised text in the Discussion section, lines 199-203, now reads: “Although the retrieval-augmented generation (RAG) framework substantially reduces the likelihood of hallucination by anchoring responses to curated clinical-genomic data, the possibility of nuanced misinterpretation for highly complex or multi-layered queries cannot be fully eliminated and should be considered when interpreting outputs.”
Comment 8:
- 8. Where feasible, merge datasets or collaborate with other groups to bolster subgroup sizes and statistical power.
Response:
We thank the reviewer for this excellent suggestion. We agree that merging datasets and collaborating with other groups would be valuable strategies to bolster subgroup sizes and increase statistical power, particularly for underrepresented alterations such as CHEK1 mutations. While this initial study focused on demonstrating the feasibility of AI-HOPE-TP53 using harmonized datasets from cBioPortal (TCGA, MSK-IMPACT, AACR GENIE), we have noted in the Discussion that future efforts will prioritize data integration across additional cohorts and collaborations with institutional and consortia-based studies. These steps will strengthen the robustness of subgroup analyses, expand representation of diverse populations, and enhance the translational utility of the platform.
The revised text in the Discussion section, lines 459-466, now reads: “AI-HOPE-TP53 integrates a fine-tuned biomedical LLM with harmonized, multi-institutional datasets from cBioPortal within a modular analytical pipeline, showcasing both technical feasibility and translational potential. The platform enables stratification across diverse clinical and demographic variables—including age, tumor site, stage, ancestry, MSI status, treatment exposure, and mutation type—positioning it as one of the few tools designed specifically for equity-focused precision oncology. Moreover, the outputs are exportable, reproducible, and visually interpretable, aligning with the core usability standards expected for Resource Reports in Cancer Research.”
The revised text in the Discussion section, lines 504-511, now reads: “While these findings demonstrate the platform’s broad utility, several limitations warrant consideration. First, because our analyses were derived exclusively from harmonized, publicly available datasets (e.g., TCGA, MSK, GENIE), the findings may not fully represent real-world populations or treatment contexts. Certain high-risk groups—including rural, Indigenous, or uninsured patients—are underrepresented in these repositories, which limits the generalizability of some disparity-related findings. For this reason, results from AI-HOPE-TP53 should be interpreted as hypothesis-generating signals requiring confirmation in larger, ancestry-diverse, and prospectively collected cohorts.”
Comment 9:
- 9. Encompass RNA-seq, proteomic, epigenomic, and spatial biology data into more comprehensive pathway interrogation. (is possible)
Response:
We thank the reviewer for this thoughtful suggestion. We agree that integrating additional data modalities such as RNA-seq, proteomic, epigenomic, and spatial biology would greatly enhance the ability of AI-HOPE-TP53 to perform more comprehensive pathway interrogation. At present, the platform is limited to DNA-level alterations, but its modular architecture is designed to accommodate expansion to multi-omic datasets. We have noted this in the Discussion, emphasizing that incorporation of transcriptomic, proteomic, epigenomic, and spatial biology data is a planned direction for future development to increase biological depth and translational relevance.
As previously mentioned, the revised text in the Discussion section, lines 516-522, now reads: “In addition, while AI-HOPE-TP53 demonstrates technical feasibility and research utility, its current clinical applicability remains limited. The platform does not yet support direct integration of institutional patient-level datasets, electronic health records (EHRs), or multi-omic layers such as RNA-seq, proteomics, and epigenomics. As such, the immediate scope of AI-HOPE-TP53 is research-focused rather than clinical decision-making. Future development will prioritize these extensions to enhance the translational and clinical utility of the platform.”
Comment 10:
- 10. Encompass RNA-seq, proteomic, epigenomic, and spatial biology data into more comprehensive pathway interrogation. (is possible)
Response:
We thank the reviewer for this excellent suggestion. We agree that incorporating additional omics layers—including RNA-seq, proteomic, epigenomic, and spatial biology data—would allow AI-HOPE-TP53 to deliver a far more comprehensive interrogation of the TP53 pathway. While the current version is limited to DNA-level alterations, the platform’s modular architecture is well-suited for expansion. We have added this point to the Discussion, noting that integration of multi-omic datasets is a priority for future development to enhance both biological insight and translational applicability.
The revised text in the Discussion section, lines 558-578, now reads: “Another limitation of this study is that validation of AI-HOPE-TP53 was restricted to the replication of known genotype–phenotype associations within publicly available datasets. While this approach establishes feasibility and reproducibility, it does not provide independent or prospective confirmation of the platform’s findings. Future development will prioritize validation using institutional cohorts, prospective clinical datasets, and external resources such as GEO and ICGC, which will allow benchmarking of AI-HOPE-TP53 against real-world populations and treatment contexts that may be underrepresented in public repositories. In addition, integration of multi-omic layers—including RNA-seq, proteomic, and epigenomic data—will enable more comprehensive pathway interrogation. These steps are essential for ensuring the generalizability and translational relevance of AI-HOPE-TP53 and represent the next phase in its roadmap toward broader clinical and research applications. Future studies will be critical to validate and expand AI-HOPE-TP53 in real-world clinical settings. Specifically, testing the platform on institutional and prospectively collected clinical-genomic cohorts will allow benchmarking against contemporary patient populations and treatment patterns. Integration of multi-omic data layers—including transcriptomics, proteomics, and epigenomics—will enhance biological interpretability and clinical precision. In addition, secure pipelines for interoperability with electronic health records (EHRs) will extend the platform’s relevance to clinical workflows. Finally, piloting AI-HOPE-TP53 across multi-institutional collaborations will provide opportunities to evaluate scalability, usability, and equity-focused impact in diverse healthcare settings.”
Comment 11:
- 11. Cross-validate against external datasets (institutional CRC cohorts, GEO, ICGC) to quantify generalizability.
Response:
We thank the reviewer for this valuable recommendation. We agree that cross-validation with external datasets—including institutional colorectal cancer (CRC) cohorts, GEO, and ICGC—would be an important step to quantify the generalizability of AI-HOPE-TP53. While the present study focused on harmonized public datasets accessed via cBioPortal (TCGA, MSK-IMPACT, AACR GENIE), we have noted in the Discussion that future work will prioritize validation across independent datasets and institutional collaborations. These efforts will help ensure that the platform’s findings are robust, reproducible, and applicable across diverse populations and clinical contexts.
As showed before, the revised text in the Discussion section, lines 558- 565, now reads: “Another limitation of this study is that validation of AI-HOPE-TP53 was restricted to the replication of known genotype–phenotype associations within publicly available datasets. While this approach establishes feasibility and reproducibility, it does not provide independent or prospective confirmation of the platform’s findings. Future development will prioritize validation using institutional cohorts, prospective clinical datasets, and external resources such as GEO and ICGC, which will allow benchmarking of AI-HOPE-TP53 against real-world populations and treatment contexts that may be underrepresented in public repositories.”
Comment 12:
- 12. Provide guidance on non-significant trend interpretation to avoid overinterpretation.
Response:
We thank the reviewer for this thoughtful suggestion. In the revised manuscript, we have added explicit guidance on how non-significant results should be interpreted. Throughout the Results, these findings are now described as trends or exploratory observations rather than definitive associations. In the Discussion, we emphasize that such outcomes are hypothesis-generating signals that require validation in larger or independent datasets. To further support cautious interpretation, we have generated a new summary Table S1, which distinguishes statistically significant findings from exploratory, non-significant trends. This addition provides readers with a clear reference to avoid overinterpretation and ensures consistency across the manuscript.
Comment 13:
- 13. Establish plans for regular dataset updates and LLM retraining to ensure continuous accuracy and relevance.
Response:
We thank the reviewer for this excellent suggestion. We agree that regular dataset updates and periodic retraining of the large language model (LLM) are essential to maintain the accuracy, reproducibility, and long-term relevance of AI-HOPE-TP53. In the revised Discussion, we now note that the platform’s modular architecture allows for straightforward integration of newly released public datasets (e.g., updated GENIE releases, TCGA legacy harmonizations) and supports retraining of the underlying LLM to incorporate emerging biomedical knowledge. We also highlight plans to establish a scheduled update cycle, ensuring that both the datasets and AI backbone remain current as the resource evolves.
The revised text in the Discussion section, lines 579-589, now reads: “To ensure the long-term sustainability and relevance of AI-HOPE-TP53, we plan to establish a framework for regular dataset updates and LLM retraining. The modular design of the platform allows for seamless integration of new releases from public resources such as AACR GENIE, TCGA harmonizations, and MSK-IMPACT expansions, thereby keeping analyses aligned with the most current data available. In parallel, the underlying large language model can be periodically fine-tuned with updated biomedical corpora to incorporate emerging scientific knowledge and evolving clinical standards. By implementing a scheduled update cycle, we aim to maintain the accuracy, reproducibility, and translational utility of AI-HOPE-TP53, ensuring that it continues to serve as a reliable and up-to-date resource for pathway-centric and disparity-aware EOCRC research.”
We are grateful to Reviewer 3 for the insightful and constructive comments, which have contributed to improving the clarity, balance, and overall rigor of the manuscript.

Round 2
Reviewer 3 Report
Comments and Suggestions for Authors
Thank you for answering questions. good luck